# Genome sequence diversity of SARS-CoV-2 obtained from clinical samples in Uzbekistan

**Alisher Abdullaev**[1]*, **Abrorjon Abdurakhimov**[1], **Zebinisa Mirakbarova**[1], **Shakhnoza Ibragimova**[1], **Vladimir Tsoy**[1], **Sharofiddin Nuriddinov**[1], **Dilbar Dalimova**[1], **Shahlo Turdikulova**[1], **Ibrokhim Abdurakhmonov**[1,2]

**1** Center for Advanced Technologies, Tashkent, Uzbekistan, **2** Center of Genomics and Bioinformatics, Academy of Sciences of Uzbekistan, Qibray Region, Tashkent, Republic of Uzbekistan

* abdullaev_alisher@yahoo.com

## Abstract

Tracking temporal and spatial genomic changes and evolution of the severe acute respiratory syndrome coronavirus 2 (SARS-CoV-2) are among the most urgent research topics worldwide, which help to elucidate the coronavirus disease 2019 (COVID-19) pathogenesis and the effect of deleterious variants. Our current study concentrates genetic diversity of SARS-CoV-2 variants in Uzbekistan and their associations with COVID-19 severity. Thirty-nine whole genome sequences (WGS) of SARS-CoV-2 isolated from PCR-positive patients from Tashkent, Uzbekistan for the period of July-August 2021, were generated and further subjected to further genomic analysis. Genome-wide annotations of clinical isolates from our study have revealed a total of 223 nucleotide-level variations including SNPs and 34 deletions at different positions throughout the entire genome of SARS-CoV-2. These changes included two novel mutations at the Nonstructural protein (Nsp) 13: A85P and Nsp12: Y479N, which were unreported previously. There were two groups of co-occurred substitution patterns: the missense mutations in the Spike (S): D614G, Open Reading Frame (ORF) 1b: P314L, Nsp3: F924, 5'UTR:C241T; Nsp3:P2046L and Nsp3:P2287S, and the synonymous mutations in the Nsp4:D2907 (C8986T), Nsp6:T3646A and Nsp14: A1918V regions, respectively. The "Nextstrain" clustered the largest number of SARS-CoV-2 strains into the Delta clade (n = 32; 82%), followed by two Alpha-originated (n = 4; 10,3%) and 20A (n = 3; 7,7%) clades. Geographically the Delta clade sample sequences were grouped into several clusters with the SARS-CoV genotypes from Russia, Denmark, USA, Egypt and Bangladesh. Phylogenetically, the Delta isolates in our study belong to the two main subclades 21A (56%) and 21J (44%). We found that females were more affected by 21A, whereas males by 21J variant ($\chi2 = 4.57$; $p \leq 0.05$, n = 32). The amino acid substitution ORF7a:P45L in the Delta isolates found to be significantly associated with disease severity. In conclusion, this study evidenced that Identified novel substitutions Nsp13: A85P and Nsp12: Y479N, have a destabilizing effect, while missense substitution ORF7a: P45L significantly associated with disease severity.

**Data Availability Statement:** The data underlying the results presented in the study are available from FlagShare Public Repository https://doi.org/10.6084/m9.figshare.19221276.v1.

**Funding:** This study has been supported by the research grant from the Ministry of Innovative Development, Republic of Uzbekistan (Research Grant number: А-ИРВ-2021-125). There was no additional external or internal funding received for this study. The funders had no role in study design, data collection and analysis, decision to publish, or preparation of the manuscript.

**Competing interests:** The authors have declared that no competing interests exist.

## Introduction

A novel coronavirus, SARS-CoV-2, was first identified in December 2019 (Wu et al., 2020) and rapidly spread around the world. According to Johns Hopkins Coronavirus Resource Center (CRC) SARS-CoV-2 infected hundreds of millions and killed more than 5 million of people until the middle of 2021 [1]. Origin of this dangerous virus is still controversial and has been a subject of interest for many researchers. Recently, analysis of genetic structure of the SARS-CoV-2 with different coronavirus genomes revealed that it has a mosaic genome and might be obtained via intragenic recombination of various virus strains [2] (Makarenkov, 2021). Scientists continue to study the SARS-CoV-2 genome and to date numerous whole genome sequence (WGS) analyses found mutational variations in the viral genome [3–6].

Currently, there a need for increased global sequencing efforts to identify the spread of new variants as quickly as possible, since analyzing genome sequencing and single-nucleotide polymorphism (SNP) calling have been a hotspot for a wide variety of epidemiological, clinical, and therapeutic efforts [3, 4, 7–9]. Although most mutations in the SARS-CoV-2 genome are expected to be either deleterious or relatively neutral, a small proportion will affect functional properties and may alter infectivity, disease severity in even after the host immunity background [3, 10, 11].

Genomic variation is informative for tracking the distribution of the virus and identifying major clades related to the various variants of SARS-CoV-2 with different epidemiological outcome [12]. The genomic variability of SARS-CoV-2 specimens scattered across the globe can underly geographically specific etiological effects. Previous studies revealed the prevalence of single nucleotide transitions as the major mutational type characterized by geographic and genomic specificity across the world [3].

Country of origin and time since the start of the pandemics were the most influential metadata associated with genomic variation. It was highlighted that some geographic regions (populations) have unusually high (many new variants) while others have low (isolated) viral phylogenetic diversity. Such studies provided a direction to prioritize genes associated with outcome predictors (e.g., health, therapeutic, and vaccine outcomes) and to improve DNA tests for predicting disease status [13]. Current study is aimed at identifying SARS-CoV-2 variants and mutation profile in the coronavirus genome presented during the second wave of coronavirus pandemic in Uzbekistan and assessing their impact on the disease severity.

Overall pattern of the COVID-19 pandemic in Uzbekistan, since the first confirmed case reported on the 15th of March, 2020, has accompanied two distinct disease waves until October of 2021. Until the 20th of October, 2021, a total of 182 060 cases and 1292 deaths were reported, of that the first peak occurred in July-August 2020, when within the two months, the total number of cases increased rapidly from 12 295 to 43 476 with two months increase of 31 181. The second waive has occurred in July to August of 2021, following the increase of infected people from 113 559 to 160 589 within the two (n = 47 030) months period (Fig 1; https://www.worldometers.info). During these two diseases waives, a number of deaths was 299 and 367, respectively, and accounted for 30.5% of total deaths (n = 2179) since pandemics started in Uzbekistan (Fig 2; https://www.worldometers.info). The lowest number of detected CVOID-19 cases in Uzbekistan (n = 1078) was observed in February 2021 (WHO, https://covid19.who.int/region/euro/country/uz).

Previous efforts [14], have characterized 18 high-quality WGS reads for SARS-COV-2 from the very early symptomatic COVID-19 patients, sampled from Tashkent region clinics for the period of October and beginning of December, 2020. After the first wave, WGS of 18 SARS-CoV-2 genomes, distributed in Uzbekistan at that period, have revealed a total of 128 SNPs, consisting of 45 shared and 83 unique mutations, phylogenetically suggesting their origin and

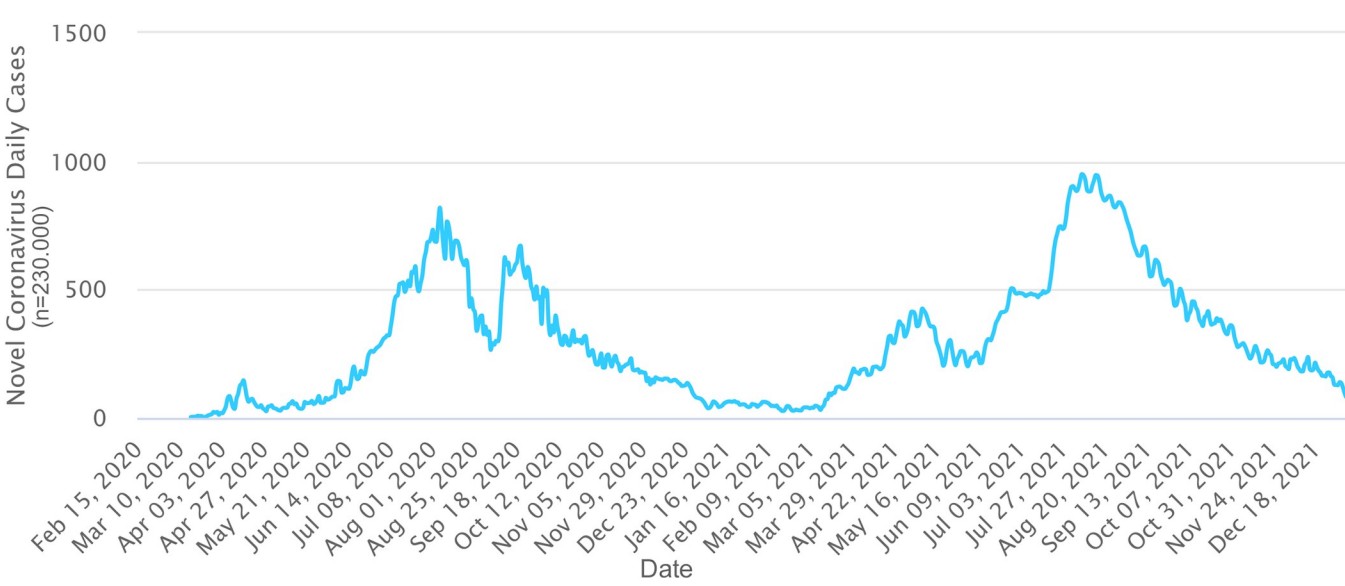

**Fig 1. COVID-19 cases in Uzbekistan since March 2020 to September 2021 (according to Worldometer, https://www.worldometers.info).**

spread from European and Near East countries because of international travels [14]. However, a periodic genome sequencing of SARS-CoV-2 should be carried out to identify the degree of mutations and novel variants of concern (VOC), potentially useful for disease diagnostics, monitoring and treatment.

Therefore, the main purpose of current study was to initiate another large-sample, WGS and genetic diversity evaluation study of the SARS-CoV-2, distributed during the second infection waive period in Uzbekistan, using virus samples isolated from COVID-19 positive

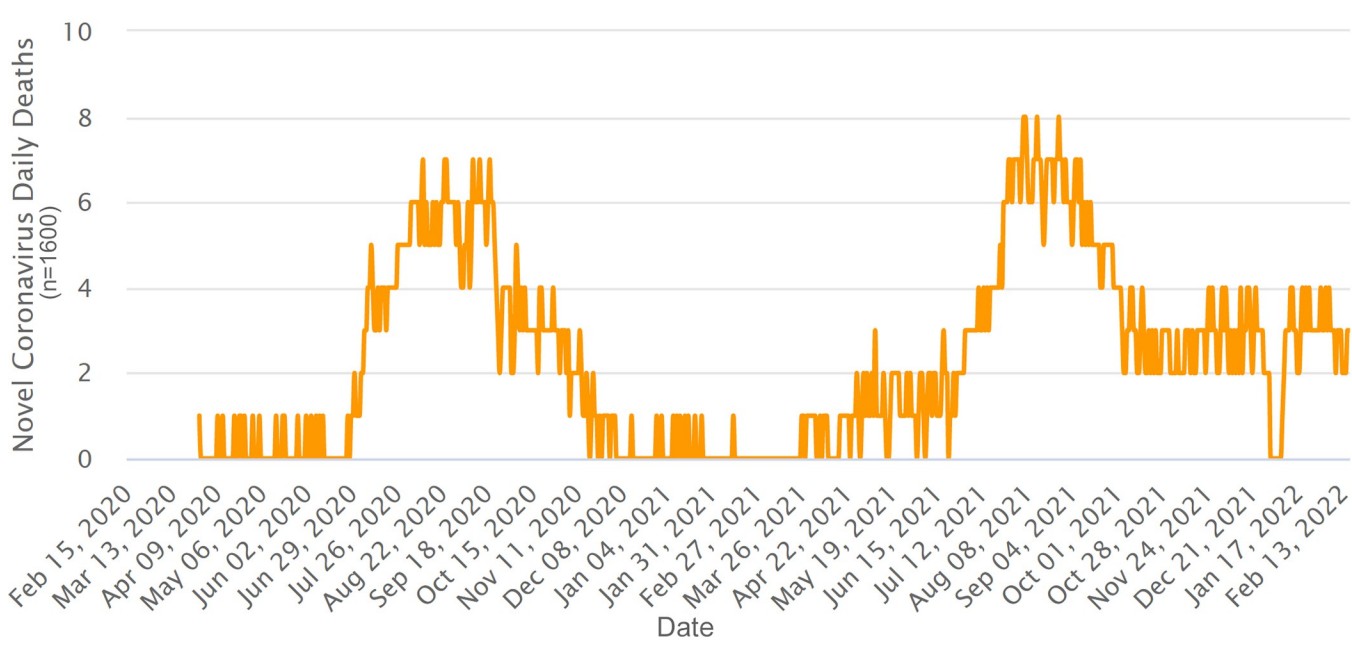

**Fig 2. Deaths caused by COVID-19 in Uzbekistan since March 2020 to September 2021 (according to Worldometer, https://www.worldometers.info).**

patients with mild, moderate, and severe symptoms. Here we report that in the second COVID-19 disease-waive, a SARS-CoV-2 has changed into several new variants with majority (82%) distribution of Delta variant in the country. We have identified 223 nucleotide-level variations and 34 deletions at different positions with the two co-occurred genomic substitution patterns and two novel mutations (Nsp13: A85P and Nsp12: Y479N). Results showed that one of missense substitutions namely ORF7a: P45L was significantly associated with disease severity.

## Materials and methods

### Samples collection

The SARS-CoV-2 strains were isolated from nasopharyngeal swab samples (Huachenyang Technology, Shenzhen, China) of SARS-CoV-2 qPCR positive (Ct≤28) patients with mild, moderate, and severe symptoms who were being treated at the State Hospital Zangiota-1 located in Tashkent, Uzbekistan, during July 2021. The study included 20 male and 19 female patients, a total of 39 samples with average ages of 55,21 (SD = 14,9; Table 1). Ethical approval for this study was obtained from Ethics Committee of Center for Advanced Technologies under Ministry of Innovative development (Approval Date: May 5, 2021; Approval Number CAT-EC-2021/05-1). Samples collected were anonymized and distinct ID numbers were assigned to each, only keeping age and biological sex for the downstream analysis and reporting purposes. Because of anonymity of the collected data, a voluntary participation condition and non-invasiveness of sequencing experiment, with clear explanation of the purpose of sample collection to each participant, we received their verbal consent. All the experiments were carried out in accordance with the relevant guidelines and regulations.

### Data collection

Epidemiological, clinical, and disease severity data were extracted from electronic medical records using a standardized data collection form, kindly provided by State Hospital Zangiota-1 located in Tashkent, Uzbekistan. All data records were matched with previously assigned sample IDs and used as anonymized dataset for downstream analyses.

### SARS-CoV-2 nucleic acid isolation

Total nucleic acid was extracted using the QIAseq DIRECT SARS-CoV-2 (Qiagen GmbH, Hilden, Germany). RNA quantity was evaluated through the Qubit RNA HS assays kits (Life Technologies, Carlsbad, California, USA) on Qubit® 2.0 Fluorometer (Life Technologies, Carlsbad, California, USA) according to manufacturer's manual. The presence of SARS-CoV-2 in the purified RNA samples for the downstream steps was estimated by fluorescence detection of RdRp and N genes using Biotest SARS-CoV-2 RT-qPCR Kit (Biotest Lab LLC, Tashkent, Uzbekistan) according to manufacturer's protocol on the QuantStudio™ 5 Real-Time PCR System (Applied Biosystems, Foster City, USA).

**Table 1. The average number of nucleotide and amino acid substitutions, and deletions per the SARS-CoV-2 genome among 39 samples identified in Uzbekistan (with respect to the reference genome NC_045512.2).**

| Mutation type | Average | SD | max | min |
|---|---|---|---|---|
| nucleotide substitutions | 31.61 | 9.234 | 45 | 9 |
| nucleotide deletions | 16.46 | 4.381 | 25 | 13 |
| amino acid substitutions | 25.46 | 7.350 | 34 | 7 |
| amino acid deletions | 5.150 | 1.463 | 8 | 4 |

## Next-generation sequencing

Whole genome amplification of the SARS-CoV-2 was performed using the CleanPlex®
SARS-CoV-2 Research and Surveillance NGS Panel (Paragon Genomics Inc., Hayward, CA,
USA). Briefly, cDNA was generated from previously extracted RNA using RT Primer Mix DP
followed by RT reaction purification from RNA using CleanMag® Magnetic Beads (Paragon
Genomics Inc., Hayward, CA, USA). Then, a multiplex PCR (mPCR) reaction using target-
specific primers to amplify the entire SARS-CoV-2 genome was performed, using a 2-pool
design, followed by digestion and post-digestion purification, and second mPCR reaction.
This was to amplify and add sample-level i5 and i7 primers into the generated libraries for the
Illumina sequencing platforms (Illumina, San Diego, CA, USA). Finally, the library was puri-
fied using CleanMag® Magnetic Beads (Paragon Genomics Inc., Hayward, CA, USA). Librar-
ies were evaluated by gel-electrophoresis and considered for sequencing when a fragment size
~ 275 bp was obtained and the final concentration was above 2.0 ng/µl, measured on Qubit™
with dsDNA HS Assay Kit (Thermo Fisher Scientific, Waltham, MA, USA).

After confirmation of the library quality, libraries were normalized to 10 nM and samples
with unique index combinations were pooled in equimolar ratios to reach the recommended
final concentration of 4 nM for sequencing. After a further quantification with Qubit dsDNA
HS Assay Kit (Thermo Fisher Scientific, Waltham, MA, USA), pooled libraries were prepared
for the sequencing, following the Standard Normalization protocol on MiSeq System Denature
and Dilute Libraries Guide (Illumina, San Diego, CA, USA) with a final denaturation and dilu-
tion to 11 pM. Sequencing was carried-out on a MiSeq instrument (Illumina, San Diego, CA,
USA) with Reagent Kit v3, using 20 pM PhiX control spike-in of 5% for low-diversity libraries,
setting 2 x 150 cycles, and generating paired-end reads.

## Genomic analyses, variants assessments and phylogeny

NGS raw data (FASTQ files) were generated from MiSeq Local Run Manager (Illumina, San
Diego, CA, USA) and uploaded on the SOPHiA DDM platform (SOPHiA Genetics, Lausanne,
Switzerland) for the external quality check, trimming of adaptors, variant call review, re-align-
ment of indels, quality measurements, and determination of the consensus genome by map-
ping to reference sequence MN908947 (NC 045512.2). For this, we have used a proprietary
design pipeline to cover the entire genome.

The public database GISAID [15] was used for the BLAST searches and for mutation analy-
sis. The Nextclade Web tool v.1.11.1 [16] was used to compare study sequences to SARS-CoV-
2 reference sequences, assign them to clades, and determine their position within the SARS-
CoV-2 phylogenetic tree.

Variant calling, and mutation identification was also additionally performed by using
Genome Detective Coronavirus Typing Tool [17], CoV-GLUE v.1.1.108 [18] and COVID-19
genome annotator (http://giorgilab.unibo.it/coronannotator/).

The full-length genomic sequences of 39 coronaviruses were aligned using the L-INS-i
method of MAFFT v7.310 [19] https://www.nature.com/articles/s41598-020-79484-8-ref-
CR23. Phylogenetic tree of 39 samples, reference genome and genomes of variants constructed
by the Neighbor-Joining method [20] was performed using MEGA X software [21]. The boot-
strap consensus tree inferred from 10000 replicates is taken to represent the evolutionary his-
tory of the taxa analyzed. Branches corresponding to partitions reproduced in less than 50%
bootstrap replicates are collapsed. The evolutionary distances were computed using the Maxi-
mum Composite Likelihood method. Codon positions included were 1st+2nd+3rd+-
Noncoding. All ambiguous positions were removed for each sequence pair (pairwise deletion

option). The phylogenetic trees were visualized by MEGA X [21] and FigTree v1.4.4 software (http://tree.bio.ed.ac.uk/software/figtree/).

## Data availability

Nucleotide sequences of SARS-CoV-2 isolates from Uzbekistan were submitted to GISAID on 2.08.2021 and available for registered users at https://www.epicov.org/, accession IDs from EPI_ISL_3188963 to EPI_ISL_3189001. The full sequence reads and genome annotation data used and/or analyzed in this study are also available in the S1 and S2 Files.

## Statistical analysis

Summary statistics and the distribution of the primary data were visualized using BoxPlotR [22]. For association analysis, patients with mild and moderate disease severity were pooled into one "non-severe" category and compared with the "severe" (severe-critical) patient group. The association between viral genotypes and disease severity was investigated using the Fisher exact test and odds ratio (OR) calculation via a 2×2 contingency table. Comparisons were made between our phylogenetic clades [e.g., a number of patients, infected with a SARS-CoV-2 from the clade 21A (or defined SNPs) versus the number of patients, not infected with a SARS-CoV-2 from the clade 21J (or defined SNPs] and the different categories of disease severity (e.g., a number of patients with severe disease versus a number of patients without severe disease).

## Protein stability prediction

Prediction the increased or decreased stability of protein upon amino-acid substitution was done by the estimation of difference in Gibbs free energy of unfolding (ΔΔG value in kcal/mol) between the mutated and wild-type proteins ($\Delta\Delta G = \Delta G_{mutant} - \Delta G_{wild\text{-}type}$) [23, 24]. Although many prediction tools are available, the conflicting prediction results from different tools could cause confusion to users. In order to avoid such bias in predicting proteins stability, several programs have been used in our analyses, which exploited the diverse prediction models from machine-learning to energy-based force-fields [25], and the average ΔΔG value was calculated for each protein.

## Results

### Genome-wide characterization of SARS-CoV-2 isolates from Uzbekistan

Forty-eight samples collected from the COVID-19 positive patients after qPCR screening were selected for further WGS. Out of 48 samples isolated from SARS-CoV-2 positive patients, we generated 39 high-quality sequences. The average genome read length of 39 samples was 29 886.5 (σ = 4.3) nucleotides, the average content of nucleotide was 29.93% of Adenine (A), 32.11% of Thymidine (T), 19.60% of Guanine (G) and 18.32% of Cytosine (C). The GC content was 37.92 percent.

Genome-wide annotations of SARS-Cov-2 positive sample isolates in our study revealed a total of 179 nucleotide-level variations, including two novel mutations, which were unreported previously, and 34 deletions at different positions throughout the entire genome of SARS-CoV-2. Comparative sequence alignment of 179 mutations to the reference genome determined that 114 (63.7%) were missense (nonsynonymous) mutations and 65 (36.3%) were silent (synonymous) mutations. The average number of SNVs per isolate was 31.6 (σ = 9.23) with minimum and maximum values of 9 and 45 respectively (Table 1; Fig 3).

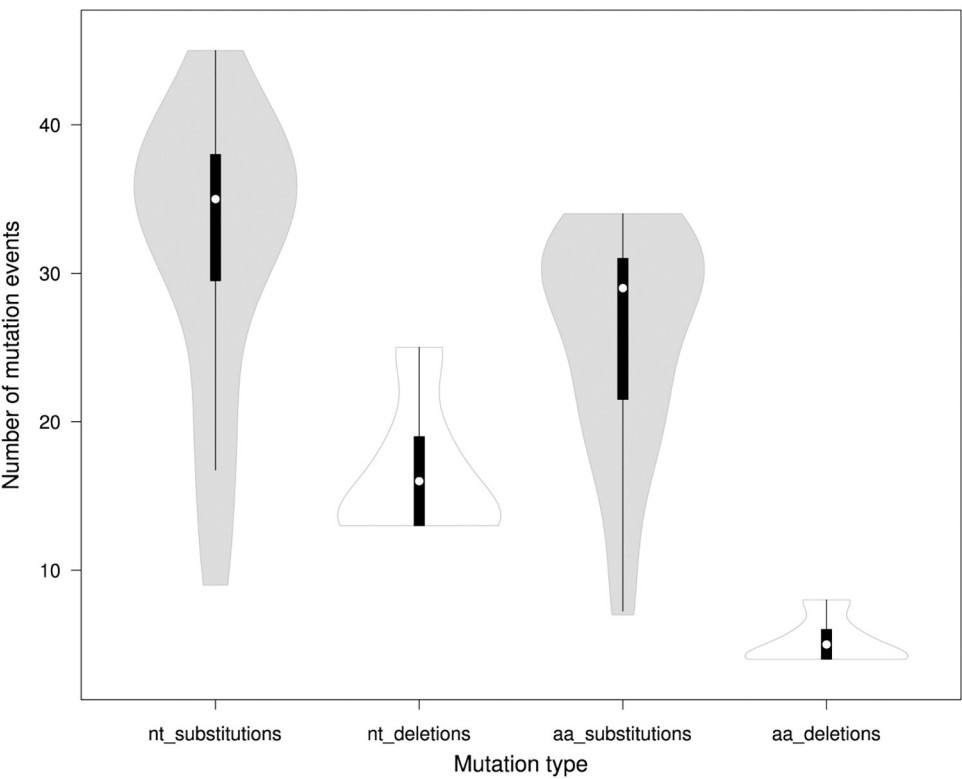

**Fig 3. Violin plot of nucleotide and amino acid substitutions, and deletions per the SARS-CoV-2 genome among 39 samples identified in Uzbekistan (with respect to the reference genome NC_045512.2).** The dots mark the median values, the bold bar marks interquartile range; nt–nucleotide; aa–amino acid.

Analysis of nucleotide substitutions rate in the SARS-CoV-2 genome showed prevalence of transitions of 68.1% with the most common C>T substitution of 42% (Figs 4 and 5; Table 2). The average number of transitions per isolate was 21. 1 (σ = 6.68), with a minimum and maximum number of 6 and 33, respectively (Fig 5). Among transversion events a G>T substitutions had the highest rate of 16% (Table 2; Fig 4).

Detailed analysis on genomic distribution of SNPs and amino acid substitutions revealed that the most variable regions are the ORF1a and ORF1b with 59 and 43 SNPs, respectively (Table 3). Interestingly, the silent SNPs were more abundant in the Nsp3 and Nsp12 (RdRp), which were nine and 10, respectively. The highest number of amino acid substitutions was found in the ORF1a (n = 34), ORF1b (n = 24) and S (24) genes (Table 3). The only one amino acid substitution I82T was present in the Membrane (M) protein (n = 29; 74% of isolates) and no mutation found in the Envelope (E) protein. Among all genes with the highest rate of amino acid substitutions was the Spike (S) gene (6 mutations with the frequency of occurrence from 79% to 100%) and the ORF1b (3 mutations with the frequency of occurrence from 77% to 97%). In the S protein, the SARS-CoV-2 mutation D614G was identified in all isolates (n = 39; 100%) from our study, followed by R158G (n = 35; 90%), T19R (n = 34; 87%), D950N (n = 33; 85%), L452R (n = 33; 85%), T478K (n = 32; 82%) and P681R (n = 31; 79%). Frequencies of all identified missense mutation in SARS-CoV-2 isolates from our study have been presented in the Table 4.

The other high frequency mutations (identified in 100% of samples) were the P314L amino acid change mutation in the ORF1b, the synonymous F924 (C3037T) change in the Nsp3 and

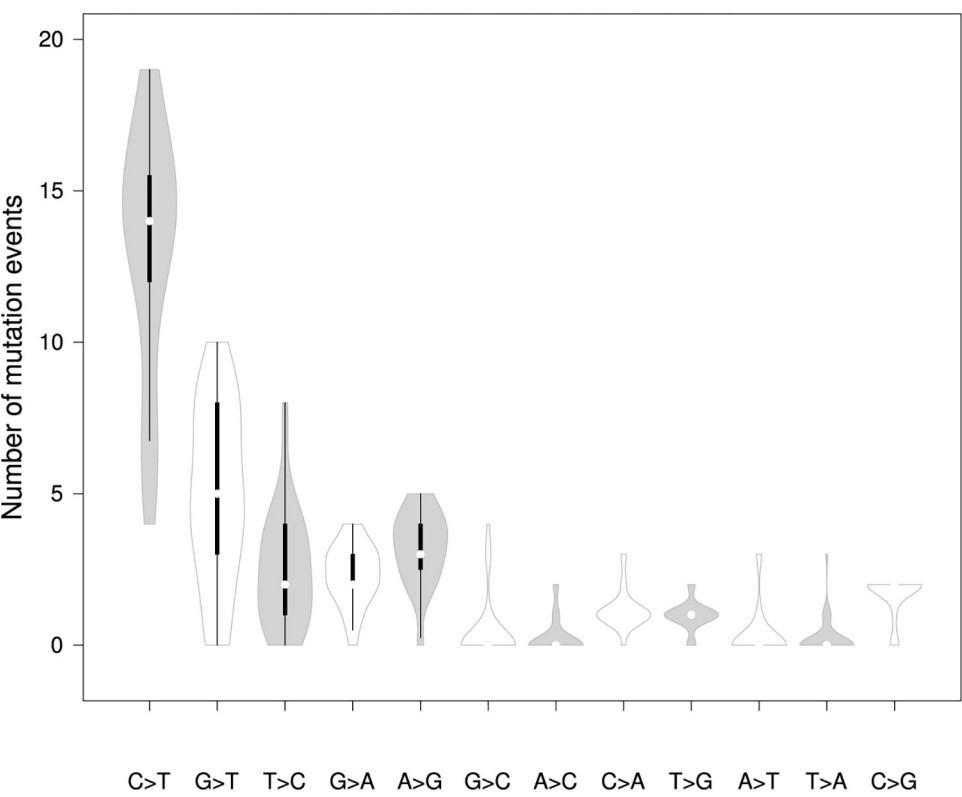

**Fig 4. Box plot of distribution of single nucleotide substitutions per genome among the SARS-CoV-2 isolates (n = 39) in Uzbekistan.** The cross marks the mean values, the bold line marks median.

the nucleotide substitution C241T in the 5'UTR (Table 4). Thus, in all isolates, regardless of their phylogenetic position, the substitution of D614G co-occurred within the ORF1b: P314L, Nsp3: F924 and 5'UTR: C241T. Further co-occurrence analysis revealed of the following 5 mutations: Nsp3: P2046L, Nsp3: P2287S, synonymous in the Nsp4: D2907 (C8986T), Nsp6: T3646A and Nsp14: A1918V.

Analysis of mutation profile among the symptomatic isolates from our study have revealed the two novel nucleotide substitutions G16489C and T14875A, which resulted in amino acid changes in the Nsp13: A85P (ORF1b: A1008P) and Nsp12: Y479N (ORF1b: Y470N), respectively (Table 4; GISAID IDs: EPI_ISL_3188967, EPI_ISL_3188979).

Genome-wide characterization of SARS-CoV-2 isolates has identified 34 nucleotide deletions resulted in 11 amino acid deletions (at the Spike, ORF8, Nsp6 and Nsp1 regions) and non-coding deletion g.a28271 at the upstream of the N gene (Table 5). This non-coding deletion g.a28271 has been identified in all SARS-CoV-2 isolates (n = 39; 100%). The average number of nucleotide deletions was 16.5 (σ = 4.38) per isolate with the minimum and maximum of 13 and 25, respectively. The average number of amino acid deletions was 5.15 (σ = 1.46) per isolate with the minimum and maximum of 4 and 8, respectively (Tables 1 and 5; Fig 3).

The frequency of amino acid deletions in non-structural proteins were 85% in the ORF8: D119/F120Δ (n = 33), 28% in the Nsp6: SGF3675-3677Δ (n = 11) and 3% in the Nsp1: M85Δ (n = 1). In the S protein there were three known amino acid deletions with the following

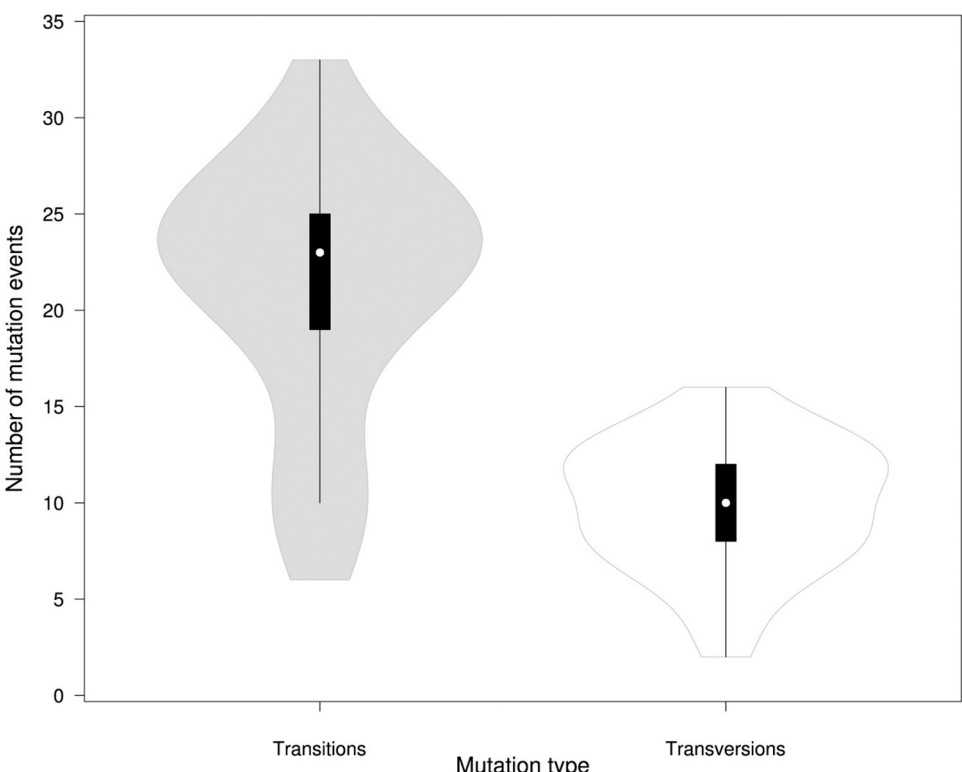

**Fig 5. Box plot of distribution of transitions and transversions per genome among the SARS-CoV-2 isolates (n = 39) in Uzbekistan.** The cross marks the mean values, the bold line marks median.

occurrence frequencies:90% in for the E156/F157Δ (n = 35; 90%), 49% for the Y144 Δ (n = 19) and 13% for the HV69-70Δ (n = 5) (Table 5).

## Phylogeny of SARS-CoV-2 isolates from Uzbekistan

The Nextclade-based phylogenetic analysis of the SARS-CoV-2 genome samples sequenced in our study has revealed that the most currently distributed SARS-CoV-2 variants in our dataset

**Table 2. Single nucleotide substitutions frequency in the genome of SARS-CoV-2 isolates (n = 39) from Uzbekistan.**

| Nucleotide substitution | Type | Average per genome | Frequency |
|---|---|---|---|
| C>T | Transition | 13.10 | 41.96% |
| G>T | Transversion | 4.90 | 15.63% |
| A>G | transition | 3.23 | 10.13% |
| T>C | transition | 2.49 | 8.14% |
| G>A | transition | 2.33 | 8.11% |
| C>G | transversion | 1.69 | 5.12% |
| C>A | transversion | 1.18 | 3.74% |
| T>G | transversion | 0.97 | 3.12% |
| G>C | transversion | 0.38 | 1.13% |
| A>T | transversion | 0.31 | 1.12% |
| A>C | transversion | 0.26 | 1.032% |
| T>A | transversion | 0.21 | 0.768% |
| | | Total: | 100% |

**Table 3. Genome distribution of the total number of identified nucleotide and amino acid substitutions among the SARS-CoV-2 (n = 39) samples from Uzbekistan.**

| Gene/region | Total number of SNVs (silent) | Total number of amino acid substitutions |
|---|---|---|
| 5'UTR | 3 | - |
| 3'UTR | 2 | - |
| ORF1a | 59 (25) | 34 |
| ORF1b | 43 (19) | 24 |
| ORF3a | 5 (0) | 5 |
| ORF7a | 6 (1) | 5 |
| ORF7b | 2 (0) | 2 |
| ORF8 | 6 (2) | 4 |
| ORF9b | 4 (1) | 3 |
| M | 3 (2) | 1 |
| N | 12 (1) | 11 |
| S | 32 (8) | 24 |

**Table 4. The frequency of missense mutation in the SARS-CoV-2 genotypes identified in Uzbekistan (nucleotide and amino acid positions are indicated relative to the reference genome NC_045512.2).**

| Gene | Gene product/region | SNP (refseq position) | AA substitution | Number of samples | Frequency |
|---|---|---|---|---|---|
| ORF1a | Nsp2 | C884T | R207C | 22 | 0.56 |
| | Nsp3 | C6402T | P2046L | 16 | 0.41 |
| | Nsp3 | C7124T | P2287S | 16 | 0.41 |
| | Nsp4 | C10029T | T3255I | 16 | 0.41 |
| | Nsp6 | A11201G | T3646A | 15 | 0.38 |
| | Nsp3 | G4181T | A1306S | 14 | 0.36 |
| | Nsp4 | G9053T | V2930L | 14 | 0.36 |
| | Nsp4 | C9891T | A3209V | 11 | 0.28 |
| | Nsp3 | C5184T | P1640L | 9 | 0.23 |
| | Nsp6 | T11418C | V3718A | 9 | 0.23 |
| | Nsp3 | C6539T | H2092Y | 8 | 0.21 |
| | Nsp6 | C11195T | L3644F | 7 | 0.18 |
| | Nsp3 | C3096T | S944L | 7 | 0.18 |
| | Nsp2 | G1048T | K261N (K81N) | 6 | 0.15 |
| | Nsp2 | C1191T | P309L | 6 | 0.15 |
| | Nsp3 | T6954C | I2230T | 4 | 0.10 |
| | Nsp3 | C3267T | T1001I | 3 | 0.08 |
| | Nsp3 | C5388A | A1708D | 2 | 0.05 |
| | Nsp3 | C5031T | T1589I | 2 | 0.05 |
| | Nsp3 | C4455T | A1397V | 1 | 0.03 |
| | Nsp6 | T11073C | F3603S | 1 | 0.03 |
| | Nsp2 | G1820A | G519S | 1 | 0.03 |
| | Nsp10 | T13188C | I4308T | 1 | 0.03 |
| | 3C-like proteinase | A10323G | K3353R | 1 | 0.03 |
| | Nsp3 | C4613T | L1450F | 1 | 0.03 |
| | Nsp2 | T1227G | M321R | 1 | 0.03 |
| | Nsp8 | T12667G | N4134K | 1 | 0.03 |
| | Nsp2 | G1161A | R299K | 1 | 0.03 |
| | Nsp3 | C5826T | T1854I | 1 | 0.03 |
| | Nsp9 | C12756T | T4164I | 1 | 0.03 |
| | Nsp3 | G6446T | V2061F | 1 | 0.03 |
| | Nsp6 | G11417A | V3718I | 1 | 0.03 |
| | Nsp8 | G12569T | V4102L | 1 | 0.03 |
| | Nsp2 | T3026A | Y921N | 1 | 0.03 |

*(Continued)*

**Table 4.** (Continued)

| Gene | Gene product/region | SNP (refseq position) | AA substitution | Number of samples | Frequency |
|---|---|---|---|---|---|
| ORF1b | ORF1ab polyprotein | C14408T | P314L | 39 | 1 |
| | | C16466T | P1000L | 33 | 0.85 |
| | | G15451A | G662S | 30 | 0.77 |
| | | C19220T | A1918V | 18 | 0.46 |
| | | C20320T | H2285Y | 6 | 0.15 |
| | | T19420C | S1985P | 4 | 0.10 |
| | | C16349T | S961L | 4 | 0.10 |
| | | A21137G | K2557R | 3 | 0.08 |
| | | A17431G | I1322V | 2 | 0.05 |
| | | C18176T | P1570L | 2 | 0.05 |
| | | G16489C† | A1008P† (NSP13: A85P) | 1 | 0.03 |
| | | G18040T | A1525S | 1 | 0.03 |
| | | G18469A | D1668N | 1 | 0.03 |
| | | G19788C | E2107D | 1 | 0.03 |
| | | T16307C | F947S | 1 | 0.03 |
| | | G16852T | G1129C | 1 | 0.03 |
| | | G21204T | K2579N | 1 | 0.03 |
| | | C17125T | L1220F | 1 | 0.03 |
| | | C14120T | P218L | 1 | 0.03 |
| | | A20059G | S2198G | 1 | 0.03 |
| | | G16741T | V1092F | 1 | 0.03 |
| | | G20578T | V2371L | 1 | 0.03 |
| | | G13726A | V87I | 1 | 0.03 |
| | | T14875A† | Y470N† (NSP12: Y479N) | 1 | 0.03 |
| 3a | ORF3a protein | C25469T | S26L | 31 | 0.79 |
| | | A25439C | K16T | 3 | 0.08 |
| | | G25947C | Q185H | 2 | 0.05 |
| | | T25520G | F43C | 1 | 0.03 |
| | | G25996T | V202L | 1 | 0.03 |
| 7a | ORF7a protein | C27752T | ORF7a:T120I | 31 | 0.79 |
| | | T27638C | ORF7a:V82A | 23 | 0.59 |
| | | C27739T | ORF7a:L116F | 11 | 0.28 |
| | | C27527T | ORF7a:P45L | 7 | 0.18 |
| | | G27478T | ORF7a:V29L | 1 | 0.03 |
| 7b | ORF7b protein | C27874T | ORF7b:T40I | 14 | 0.35 |
| | | T27835C | ORF7b:I27T | 1 | 0.03 |
| 8 | ORF8 protein | A28095T | ORF8:K68* | 4 | 0.10 |
| | | A28111G | ORF8:Y73C | 4 | 0.10 |
| | | C27972T | ORF8:Q27* | 3 | 0.08 |
| | | G28048T | ORF8:R52I | 1 | 0.03 |
| 9b | ORF9b protein | A28461G | ORF9b:T60A | 33 | |
| | | C28291T | ORF9b:P3L | 4 | 0.10 |
| | | G28396A | ORF9b:G38D | 1 | 0.03 |
| M | Membrane glycoprotein | T26767C | I82T | 29 | 0.74 |
| N | Nucleocapsid phosphoprotein | | | | |
| | NTD: RBD | A28461G | D63G | 33 | 0.85 |
| | C-tail | G29402T | D377Y | 32 | 0.82 |
| | SR-R | G28881T | R203M | 30 | 0.77 |
| | LQ-R | G28916T | G215C | 15 | 0.38 |
| | C-tail | G29427A | R385K | 9 | 0.23 |
| | N-tail | G28280C A28281T T28282A | D3L | 4 | 0.10 |
| | LQ-R | C28977T | S235F | 4 | 0.10 |
| | CTD | C29358T | T362I | 4 | 0.10 |
| | SR-R | G28883C | G204R | 3 | 0.08 |
| | SR-R | G28881A, G28882A | R203K | 3 | 0.08 |
| | NTD | G28739T | A156S | 1 | 0.03 |
| | LQ-R | G28975A | M234I | 1 | 0.03 |

(*Continued*)

**Table 4.** (Continued)

| Gene | Gene product/region | SNP (refseq position) | AA substitution | Number of samples | Frequency |
|---|---|---|---|---|---|
| S | Spike protein | | | | |
| | S1: SD2 | A23403G | D614G | 39 | 1.00 |
| | S1: NTD | del22029-22034 | R158G | 35 | 0.90 |
| | S1: NTD | C21618G | T19R | 34 | 0.87 |
| | S2: HR1 | G24410A | D950N | 33 | 0.85 |
| | S1: RBD: RBM | T22917G | L452R | 33 | 0.85 |
| | S1: RBD: RBM | C22995A | T478K | 32 | 0.82 |
| | S1: SD2 | C23604G | P681R | 31 | 0.79 |
| | S1: RBD: RBM | A23063T | N501Y | 7 | 0.18 |
| | S1: NTD | C21846T | T95I | 7 | 0.18 |
| | S1: NTD | G21987A | G142D | 5 | 0.13 |
| | S1: SD1 | C23271A | A570D | 4 | 0.10 |
| | S2 | G24914C | D1118H | 4 | 0.10 |
| | S2 | A24110C | I850L | 4 | 0.10 |
| | S1: SD2 | C23604A | P681H | 3 | 0.08 |
| | S2: HR1 | T24506G | S982A | 3 | 0.08 |
| | S2 | C23709T | T716I | 3 | 0.08 |
| | S1 | A22320C | D253A | 1 | 0.03 |
| | S2: TM | G25229A | G1223S | 1 | 0.03 |
| | S2 | T24903C | I1114T | 1 | 0.03 |
| | S1: RBD | A22814G | I418V | 1 | 0.03 |
| | S2 | G24751T | L1063F | 1 | 0.03 |
| | S1: NTD | C22323T | S254F | 1 | 0.03 |
| | S1: SD2 | T23600C | S680P | 1 | 0.03 |
| | S2: TM | G25250T | V1230L | 1 | 0.03 |

Nsp–nonstructural protein; ORF–open reading frame; S1, S2—Spike subunits 1 and 2; NTD—N-terminal domain; RBD—Receptor Binding Domain; RBM–Receptor Binding Motif; SD1, SD2—the subdomains 1 and 2; HR1- heptad repeats; TM—transmembrane domain; SR-R–Serine/Arginine-rich region; LQ-R—Leucin/Glutamine-rich region; CTD—C-terminal domain

*–nonsense mutation

†-novel, unreported substitutions

belong to the tree clades: 20A (Alpha, V1), 20I (Alpha, V1), and 21A (Delta) in consonance with "Nextstrain"'s nomenclature system [16]. Neighbor Joining analysis and the "Nextstrain" classification has grouped 82% of our SARS-CoV-2 sample genome sequences into the Delta variant clade (n = 32; 82%), followed by Alpha (n = 4; 10.3%) and 20A (n = 3; 7.7%) clades (Figs 6 and 7).

**Table 5. The frequency of deletions identified among the SARS-CoV-2 isolates (n = 39) in Uzbekistan.**

| nt deletion | Genomic region | aa deletion | Samples | Freq |
|---|---|---|---|---|
| g.a28271- | noncoding, upstream N gene | - | 39 | 1.00 |
| 22029–22034 | S | EF156-157Δ | 35 | 0.90 |
| 28248–28253 | ORF8 | DF119-120Δ | 33 | 0.85 |
| 21992–21994 | S | Y144Δ | 19 | 0.49 |
| 11288–11296 | NSP6 | SGF 3675–3677Δ | 11 | 0.28 |
| 21765–21770 | S | HV69-70Δ | 5 | 0.13 |
| 516–518 | NSP1 | M85Δ | 1 | 0.03 |

Positions are indicated relative to the reference genome (NC_045512.2)

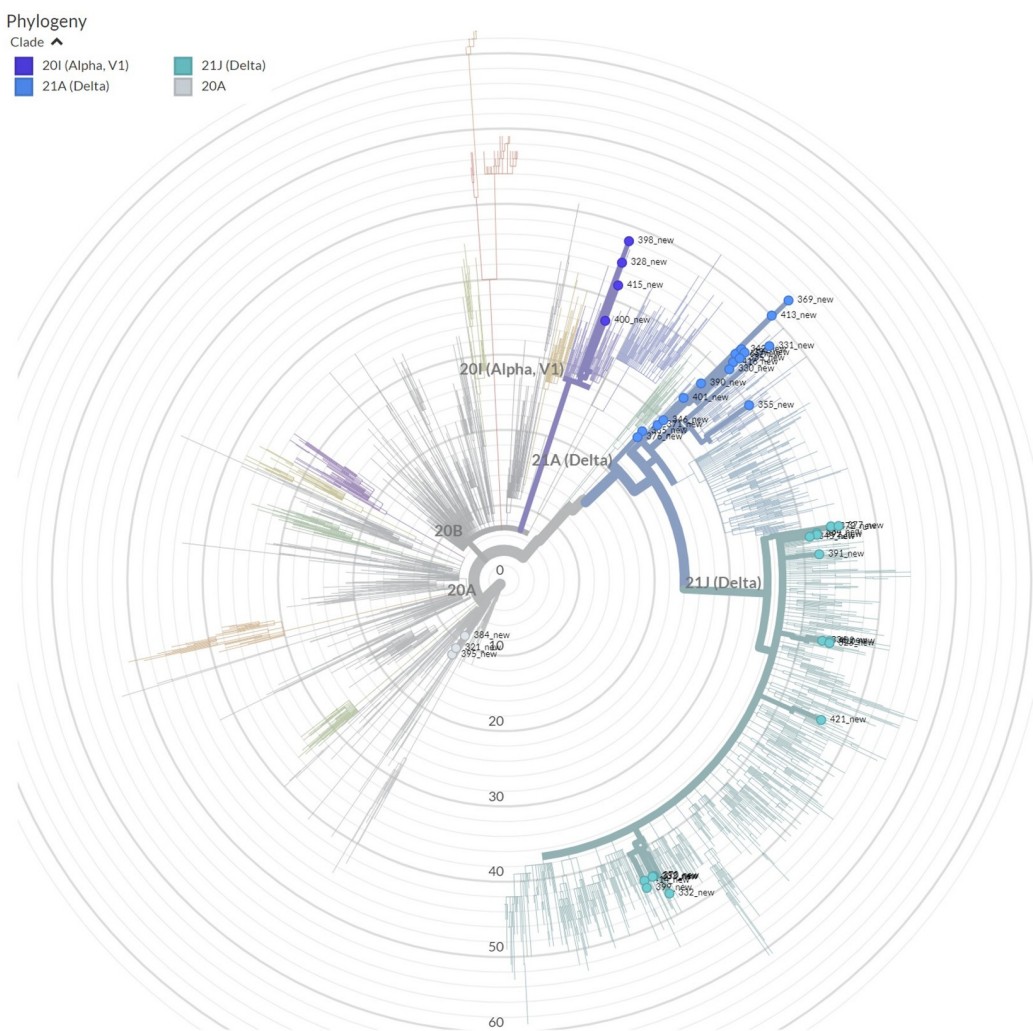

**Fig 6. The Nextclade based phylogenetic tree of the SARS-CoV-2 variants isolated in Uzbekistan.** Sequences are placed on a reference tree, clades were assigned to the nearest neighbor, and private mutations analyzed. Brunches with colored circle represents variants from Uzbekistan.

The Nexstclade phylogenetic analysis based on the WGS showed that Alpha clade variants from our dataset, namely EPI_ISL_3188994 and EPI_ISL_3188999, have clustered with the SARS-COv-2 genome sequences from France and Armenia, whereas EPI_ISL_3188965 and EPI_ISL_3188992 from our study have clustered with England coronavirus sample sequences. The clade 20A sequences from our dataset such as EPI_ISL_3188963, EPI_ISL_3188985 and EPI_ISL_3188990 clustered with the SARS-CoV-2 variants sequenced from USA and England. Our Delta clade grouped sequence reads were clustered with coronavirus sample sequence reads from Russia, Denmark, USA, Egypt and Bangladesh.

## Phylogeny and genomic characterization of the Delta isolates from Uzbekistan

Since most of the isolates were classified to the Delta clade, further step was to differentiate them by variations in the S protein sequence and other genomic regions. Analysis of the S gene mutation profile among Delta variants from our study has revealed a total 31 mutations,

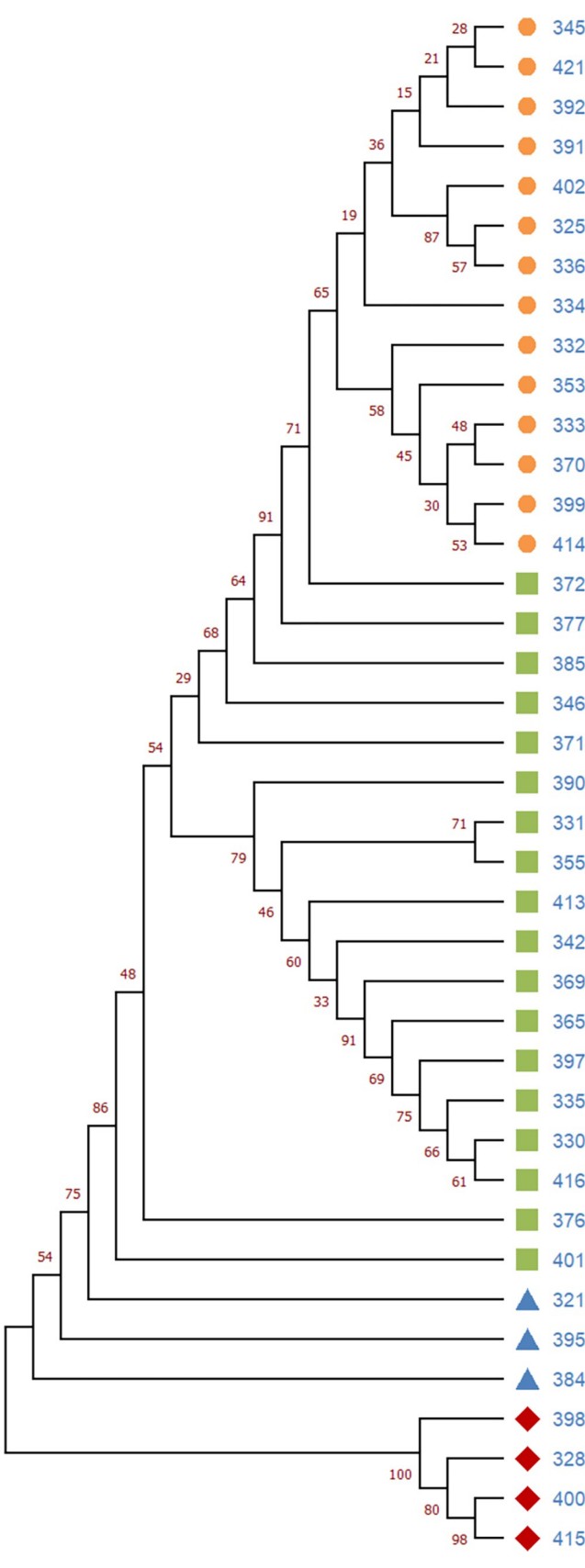

**Fig 7. Neighbor-joining phylogenetic tree of the SARS-CoV-2 isolates from Uzbekistan.** Samples are colored by taxonomic affiliation to clades or subclades. Clades are assigned according to Nextclade nomenclature. Delta variant divided to subclades 21A (green) and 21J (orange). Blue color—clade 20A, Red color–clade 20I (Alpha variant). Bootstrap values were shown.

including five amino acid deletions and eight synonymous SNP mutations. In the Spike protein, high frequency mutations, covering the 94–100% of samples sequenced, were T19R, EFR156-158G, L452R, T478K, D614G, P681R and D950N (Table 6). Along with above variants, there were several genotypes that also have contained Y144Δ (38%), T95I (22%), G142D (16%) and I850L (13%) variations in the Spike protein (Table 6). The Delta genotypes sampled from Uzbekistan in our study were also characterized by occurrence of the high frequency (91%-100%) of nonsynonymous changes in other genomic regions, namely at the ORF1b: P314L, G662S and P1000L, ORF3a: S26L, ORF7a: T120I ORF9b: T60A, N: D63G, N: D377Y, N: R203M, and M: I82T (Table 7).

**Table 6. Frequency of the identified Spike protein mutations (missense/synonymous) and deletions in the Delta isolates from Uzbekistan.**

| aa* position | Reference | Variant | nt position | mutation | total | Freq | Type |
|---|---|---|---|---|---|---|---|
| 19 | T | R | 21618 | T>G | 32 | 1.00 | missense |
| 69 | H | ΔH | 21765–21768 | Δ | 1 | 0.03 | deletion |
| 70 | V | ΔV | 21769–21770 | Δ | 1 | 0.03 | deletion |
| 95 | T | I | 21618 | C>G | 7 | 0.22 | missense |
| 142 | G | D | 21987 | G>A | 5 | 0.16 | missense |
| 144 | Y | ΔY | 21992–21994 | Δ | 12 | 0.38 | deletion |
| 156 | E | ΔE | 22029–22030 | Δ | 32 | 1.00 | deletion |
| 157 | F | ΔF | 22031–22033 | Δ | 32 | 1.00 | deletion |
| 158 | R | G | 22034 | Δ | 32 | 1.00 | missense |
| 253 | D | A | 22320 | A>C | 1 | 0.03 | missense |
| 297 | S | S | 22453 | A>C | 1 | 0.03 | synonymous |
| 302 | T | T | 22468 | G>T | 1 | 0.03 | synonymous |
| 312 | I | I | 22498 | C>T | 1 | 0.03 | synonymous |
| 375 | S | S | 22687 | C>A | 2 | 0.06 | synonymous |
| 418 | I | V | 22814 | A>G | 1 | 0.03 | missense |
| 452 | L | R | 22917 | T>G | 30 | 0.94 | missense |
| 478 | T | K | 22995 | C>A | 31 | 0.97 | missense |
| 501 | N | Y | 23063 | A>T | 1 | 0.03 | missense |
| 614 | D | G | 23403 | A>G | 32 | 1.00 | missense |
| 665 | P | P | 23557 | A>T | 1 | 0.03 | synonymous |
| 680 | S | P | 23600 | T>C | 1 | 0.03 | missense |
| 681 | P | R | 23604 | C>G | 31 | 0.97 | missense |
| 808 | D | D | 23986 | T>C | 1 | 0.03 | synonymous |
| 850 | I | L | 24110 | A>C | 4 | 0.13 | missense |
| 950 | D | N | 24410 | G>A | 32 | 1.00 | missense |
| 982 | S | A | 24506 | T>G | 1 | 0.03 | missense |
| 1061 | V | V | 24745 | C>T | 8 | 0.25 | synonymous |
| 1063 | L | F | 24751 | G>T | 1 | 0.03 | missense |
| 1223 | G | S | 25229 | G>A | 1 | 0.03 | missense |
| 1230 | V | L | 25250 | G>T | 1 | 0.03 | missense |
| 1238 | T | L | 25276 | C>T | 1 | 0.03 | synonymous |

*-aa–amino acid, positions are indicated relative to the reference genome (NC_045512.2)

**Table 7. Mutations in the Delta isolates (21A and 21J) with the frequency of occurrence higher than 20%.**

| nt position* | variant | total | frequency |
|---|---|---|---|
| A23403G | S: D614G | 32 | 1.00 |
| C14408T | ORF1b: P314L | 32 | 1.00 |
| del22029-22034 | S: R158G | 32 | 1.00 |
| C21618G | S: T19R | 32 | 1.00 |
| A28461G | N: D63G | 32 | 1.00 |
| A28461G | ORF9b: T60A | 32 | 1.00 |
| G24410A | S: D950N | 32 | 1.00 |
| G29402T | N: D377Y | 32 | 1.00 |
| C16466T | ORF1b: P1000L | 31 | 0.97 |
| C22995A | S: T478K | 31 | 0.97 |
| C23604G | S: P681R | 31 | 0.97 |
| C25469T | ORF3a: S26L | 31 | 0.97 |
| C27752T | ORF7a: T120I | 31 | 0.97 |
| T22917G | S: L452R | 30 | 0.94 |
| G28881T | N: R203M | 29 | 0.91 |
| G15451A | ORF1b: G662S | 29 | 0.91 |
| T26767C | M: I82T | 29 | 0.91 |
| T27638C | ORF7a: V82A | 23 | 0.72 |
| C884T | ORF1a: R207C | 18 | 0.56 |
| C19220T | ORF1b: A1918V | 18 | 0.56 |
| C6402T | ORF1a: P2046L | 16 | 0.50 |
| C7124T | ORF1a: P2287S | 16 | 0.50 |
| C10029T | ORF1a: T3255I | 16 | 0.50 |
| G28916T | N: G215C | 16 | 0.50 |
| A11201G | ORF1a: T3646A | 16 | 0.50 |
| G4181T | ORF1a: A1306S | 14 | 0.44 |
| G9053T | ORF1a: V2930L | 14 | 0.44 |
| C27874T | ORF7b: T40I | 14 | 0.44 |
| C9891T | ORF1a: A3209V | 11 | 0.34 |
| C27739T | ORF7a: L116F | 11 | 0.34 |
| G29427A | N: R385K | 9 | 0.28 |
| C5184T | ORF1a: P1640L | 9 | 0.28 |
| T11418C | ORF1a: V3718A | 9 | 0.28 |
| C6539T | ORF1a: H2092Y | 8 | 0.25 |
| C11195T | ORF1a: L3644F | 7 | 0.22 |
| C3096T | ORF1a: S944L | 7 | 0.22 |
| C27527T | ORF7a: P45L | 7 | 0.22 |
| C21846T | S: T95I | 7 | 0.22 |

*-nt–nucleotide, positions are indicated relative to the reference genome

According to the Nexststrain phylogenetic grouping based on the amino acid and several nucleotide substitutions, the Delta isolates from our study were divided into the two main sub-clades 21A (n = 18; 56.2%) and 21J (n = 14; 43.8 (Figs 6 and 7). All isolates in the subclade 21J were mainly defined by a frequently observed synonymous nucleotide substitution C8986T, and the amino acid substitutions at the ORF1a: A1306S (G4181T), P2046L (C6402T), P2287S (C7124T), V2930L (G9053T), T3255I (C10029T), T3646A (A11201G), ORF1b: A1918V

**Table 8. Sample mean values of nucleotide and amino acid mutation type between the subclades 21A and 21J.**

| Types | 21J (n = 14) | $\sigma^2$ | 21A (n = 18) | $\sigma^2$ |
|---|---|---|---|---|
| nt substitutions | 36.57 (SD = 2.31) | 4.96 | 30.56 (SD = 8.94) | 75.58 |
| nt deletions | 14.29 (SD = 32) | 9.92 | 17.00 (SD = 4.93) | 23.48 |
| aa substitutions | 31.36 (SD = 1.86) | 3.23 | 24.11 (SD = 5.69) | 30.65 |
| aa deletions | 4.43(SD = 1.08) | 1.10 | 5.30 (SD = 1.64) | 2.55 |

(C19220T), and ORF7b: T40I (C27874T). Three of them (ORF1a: A1306S, V2930L, and ORF1b: A1918V) were unique to the 21J subclade, while the rest were present of subclade 21A, formed by two (11%) of sample genomes sequenced. In the subclade 21J, there were also 50% of isolates bearing the S protein—specific mutation T95I (C27874T).

Isolates in the subclade 21J were characterized by a higher number of average nucleotide and amino acids substitutions, but lower nucleotide and amino acids deletions per isolate compared to the subclade 21A (Table 8, Fig 8). The broader variation range of mutations events was observed among isolates in the subclade 21A (Table 8, Fig 8) than the subclade 21J. It was found that the substitutions N: R385K, ORF1a: S944L, H2092Y, L3644F, and ORF1b: H2285Y were always co-occurred and present in 50% of isolates in the subclade 21A of our study. The Nextclade based analysis of geographical distribution indicated that all samples in the subclade

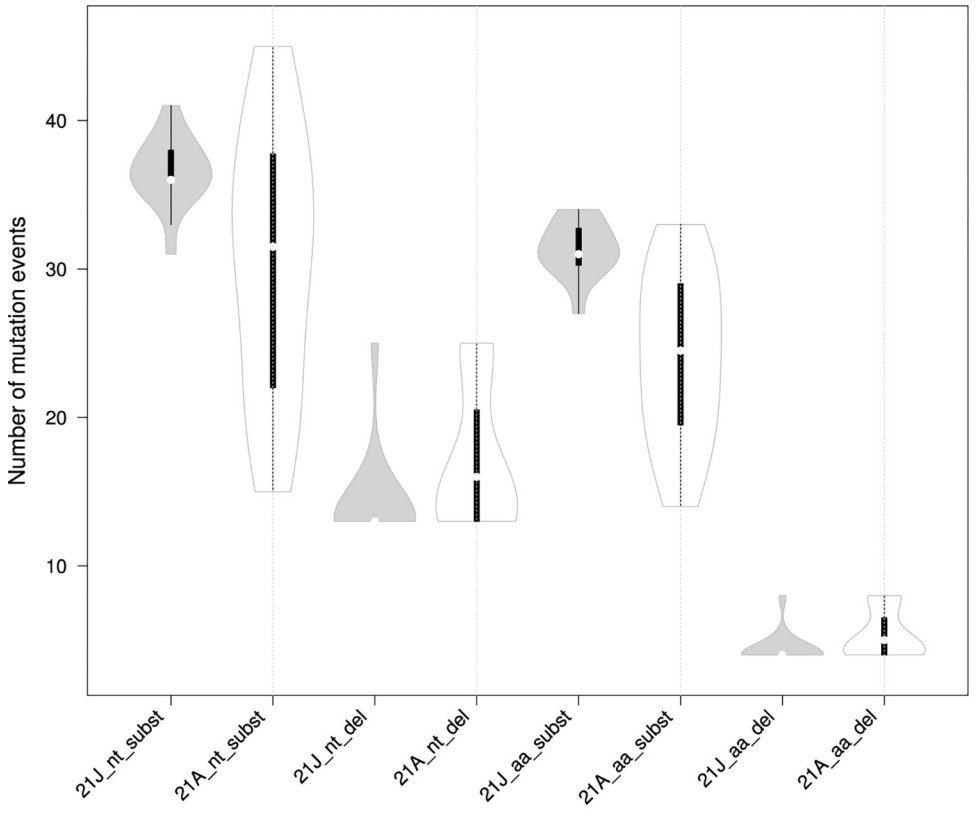

**Fig 8. Distribution of mutation types in the Delta variant subclades 21A (n = 18) and 21J (n = 14).** The dots mark the median values, the bold bar marks interquartile range. nt–nucleotide; aa–amino acid.

**Table 9. Age and gender distribution of studied patients.**

|  | Mean Age (SD) | Number of patients |
|---|---|---|
| Male | 50 (14,7) | 20 |
| Female | 60.7 (13,2) | 19 |
| Total (M+F) | 55.21 (14,9) | 39 |

21A were mostly clustered close to the SARS-CoV-2 genotypes from Bangladesh, whereas all samples in the subclade 21J were mostly clustered with the genome sequence samples of coronavirus from Egypt and Russia.

## Analysis of disease severity among patients infected with the Delta variant

The gender distribution of selected patients in our study was 51.3% (n = 20) of males, and 48.7% (n = 19) of females, with age distribution of 50 (SD = 14.7) and 60.7 (SD = 13.2) years respectively, the mean age of all 39 patients in both gender groups was 55.2 (SD = 14.9) (Table 9; Fig 9).

A number of patients with mild, severe, and critical manifestation of COVID-19 were 46.2% (n = 18), 33.3% (n = 13), and 20.5% (n = 8) respectively. No statistically significant associations were found between disease severity vs. clades/subclades (p = 0.32), and between disease severity vs. gender (p = 0.4).

Clinical symptoms among patients affected by the Delta variant (n = 32) were distributed by the disease severity to three groups: mild 46.9% (n = 15), severe 31.2% (n = 10), critical 21.9% (n = 7), and death (2.5% of all studied). Almost all hospitalized patients in three groups had at least two chronic comorbidities, including obesity, cardiovascular diseases, and diabetes. Examining the effect of comorbidities on a clinical outcome, there were no significant associations found among the above-mentioned three groups related to comorbidities and disease severity. Interestingly, a distribution analysis of the Delta subclades among gender groups revealed that females were more affected by the 21A, whereas males by the 21J variants (OR = 4.73; 95% CI 1.05–24.44; $\chi 2$ = 4.57; p = 0.032; Fig 10). There were no significant differences found between gender and disease severity (p = 0.4).

Association analysis of different amino acid substitutions, including the two novel Nsp13:A85P and Nsp12:Y479N mutations, no mutation occurred within 39 SARS-CoV-2 genomes have revealed a significant association with disease severity, except for P45L (C27527T) in the ORF7a region. Patients, infected with a SARS-CoV-2 bearing the ORF7a 45P variant found in our study, have developed severe and critical form of the COVID-19. The SARS-CoV-2 bearing 45L amino acid change in the ORF7a was predominantly found in patients with mild symptoms. To determine whether the ORF7a:P45L substitution associated with mild or severe and critical disease symptoms wused logistic regression analysis and found that the ORF7a:45L has been significantly associated with causing the mild disease symptoms (CMILE OR = 0.1; 95% CI 0.01–0.67; p<0,01), while the ORF7a:45P has been significantly associated with the severe and critical COVID-19 symptoms (CMILE OR = 9.1; 95% CI 1.38–84.98; p<0,01) (Table 10).

## Effect of amino acid substitution in the ORF7a: P45L as well as novel variants of the Nsp13:A85P, and Nsp12: Y479N on protein stability

To measure protein stability of mutated variants of SARAS-CoV-2 sequenced in our study, we have analyzed the variability of Gibbs free energy of unfolding ($\Delta\Delta G$) values. We used the diverse prediction models of machine-learning to energy-based force-fields with different

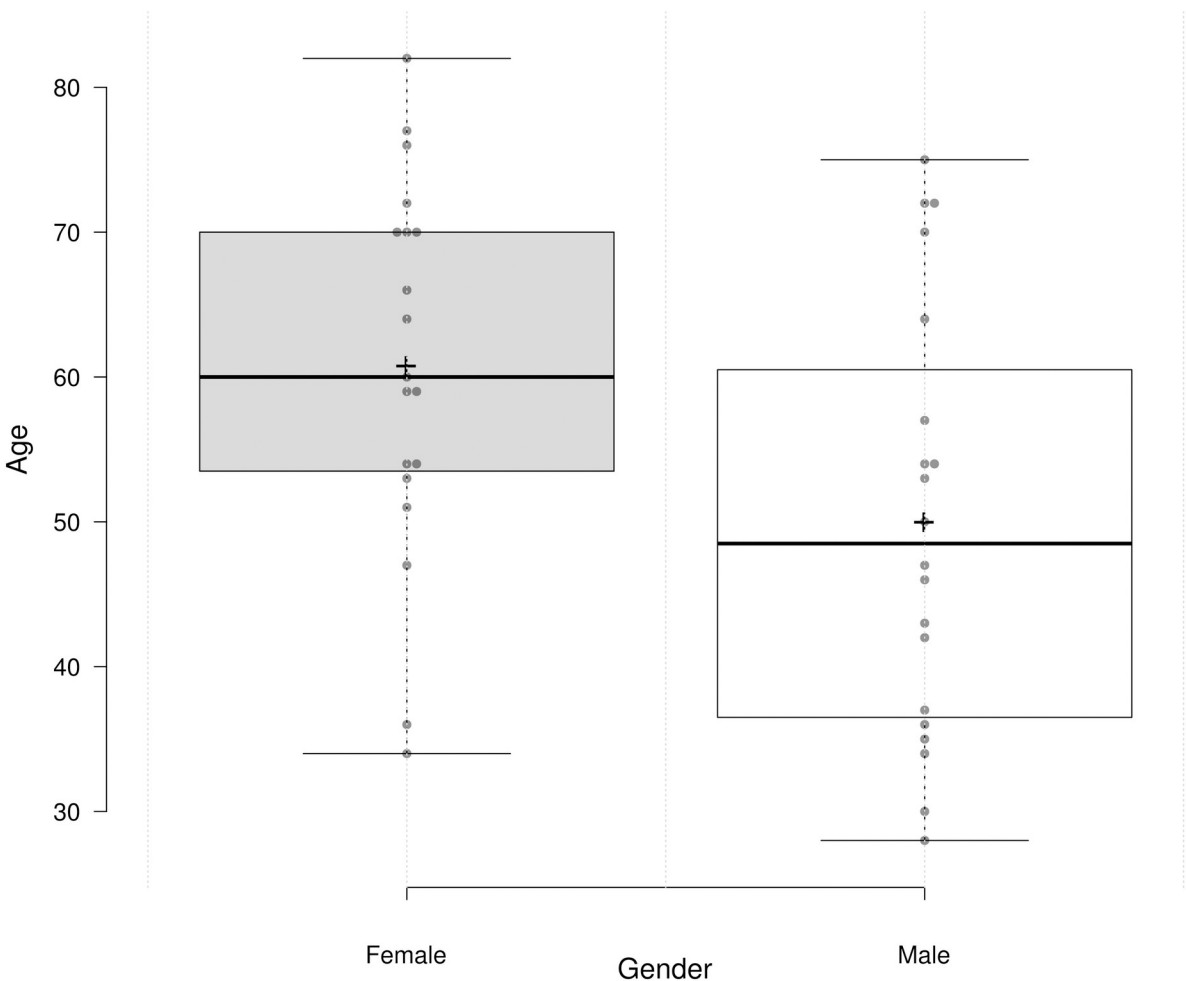

**Fig 9. Age and gender distribution among COVID-19 patients.**

software programs (Fig 11). Analysis of average ΔΔG values, in terms of increased or decreased stability of the ORF7a: P45L and novel variants of the Nsp13: A85P, and Nsp12: Y479N, has revealed a significant decrease of protein stability with an average ΔΔG values of -0.48 (s = 0.51), -0.56 (s = 1.08), and -1.5; (s = 1.26), respectively (Fig 11). The P45L substitution revealed that non-synonymous Pro to Leu substitution has a deleterious effect, leading to a highly decreased stability of theORF7a protein (Fig 12). Thus, our finding suggests that non—synonymous substitution of amino acid proline to leucine in the position 45 of the ORF7a protein strongly affects to the functional activity of protein and weakening disease symptoms in comparison with the original coronavirus protein.

## Discussion

The of the number of individuals affected by COVID-19 depends on several factors which have had an impact on whether new COVID-19 cases are increasing or declining in particular locations. These factors include the infection prevention policies, human behavior, effectiveness of vaccines over time, changes to the coronavirus itself, and the number of people in population who are vulnerable because of many reasons, including the age, genetic and immune status of hosts, and other social aspects of epidemy.

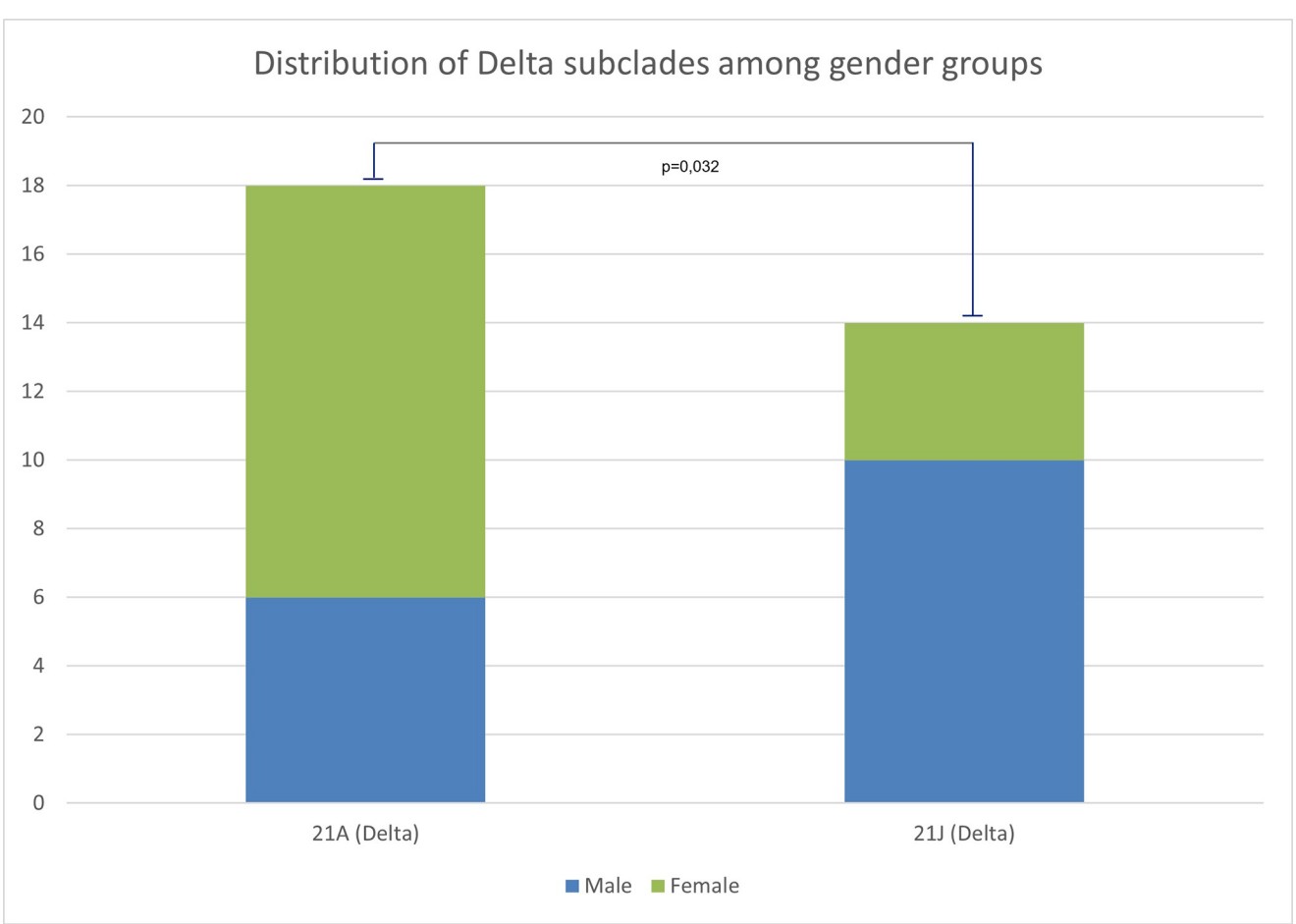

**Fig 10. Distribution of SARS-CoV-2 Delta subclades between male and female patients in Uzbekistan.**

The study of the global situation showed that the COVID-19 disease pandemic in different countries had a various number of upward periods (incidence peaks), which varied from zero to five waives/incidence, among them the largest number of countries fell in two incidence peaks [27]. Our study also shows that in Uzbekistan, for the entire period of the COVID_19 disease pandemic, two clear waves were identified (Fig 1). The first wave was thought to be provoked by the spread of the original Wuhan variant [28] in middle of 2020. However, a previous SARS-CoV-2 sample sequencing effort from Uzbekistan, covering the first waive period (November to December of 2020), concluded that the early SARS-CoV-2 infections in our country were distributed from European and Near East countries as a result of international travelling [14] and represented by clades 20B (77,7%) and 19B (22,3%), whereas during the

**Table 10. Logistic regression analysis of viral ORF7a:P45L mutation and disease severity.**

| Mutation | CMLE OR | Lower 95% CI | Upper 95% CI | p-value | p-value (F) |
|---|---|---|---|---|---|
| ORF7a: 45L | 0.1* | 0.01 | 0.67 | 0.004 | 0.03** |
| ORF7a: 45P | 9.1* | 1.38 | 84.98 | 0.004 | 0.03** |

*Conditional maximum likelihood estimates of Odds Ratio.

** Fisher's exact test two-tailed p-value

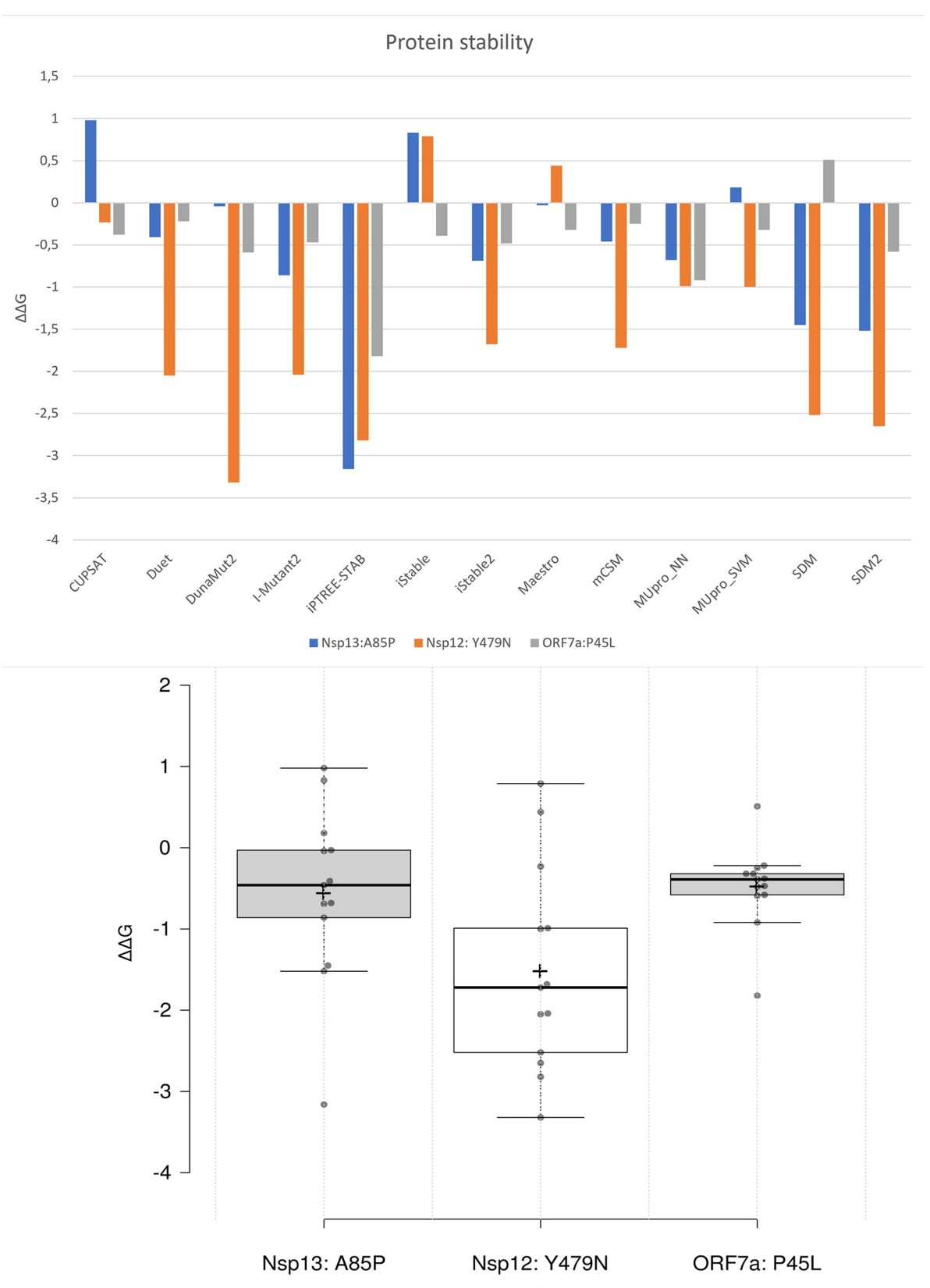

**Fig 11. Analysis of different prediction models of the NSP12: Y479N, NSP13: A85P and ORF7a: P45L amino-acid substitution on protein stability.** ΔΔG—free energy of unfolding (kcal/mol). The negative and positive predicted ΔΔG values mean the destabilizing and stabilizing effect, respectively [24, 26].

second wave dominant variant was Delta (82%), followed by Alpha (10.3%) and 20A (7.7%) clades (Figs 6 and 7), comparative phylogenetic tree between SARS-CoV-2 isolates from two waves presented in Fig 13.

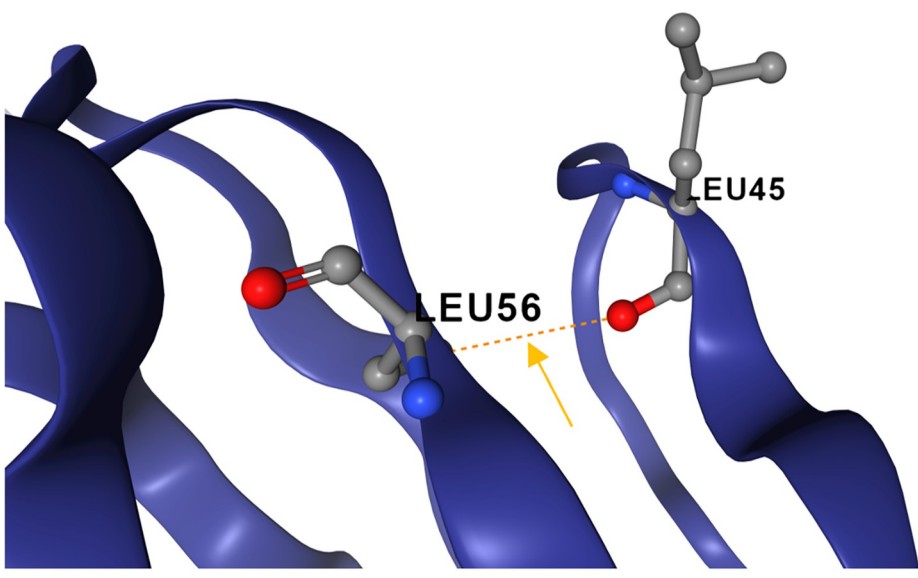

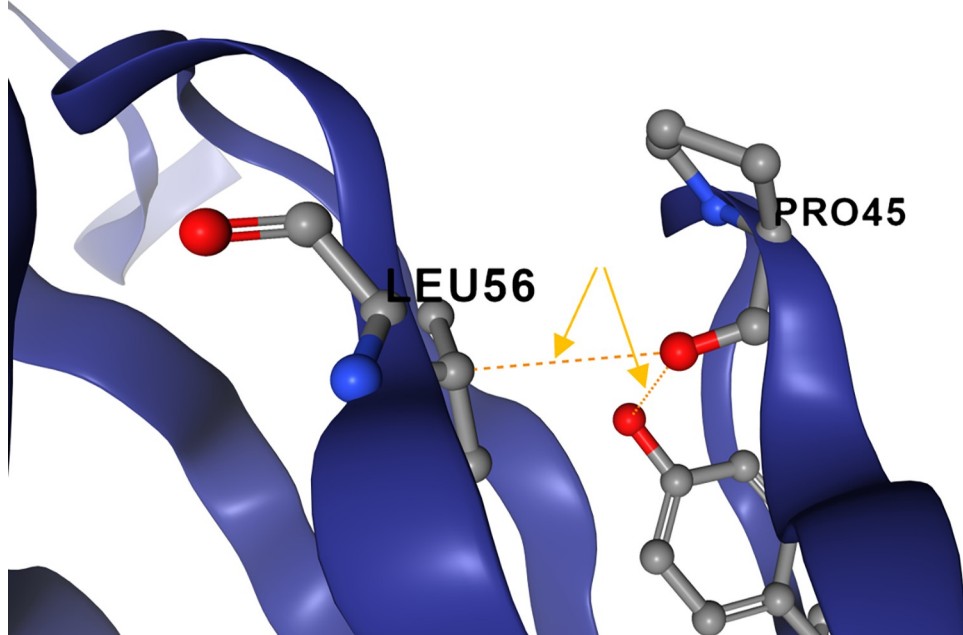

**Fig 12. 3D structure of the ORF7a: P45L substitution effect on protein stability.** Dotted line represents hydrogen bonds. Substitution of Pro to Leu (left) results in destabilizing effect due to loss of hydrogen bond.

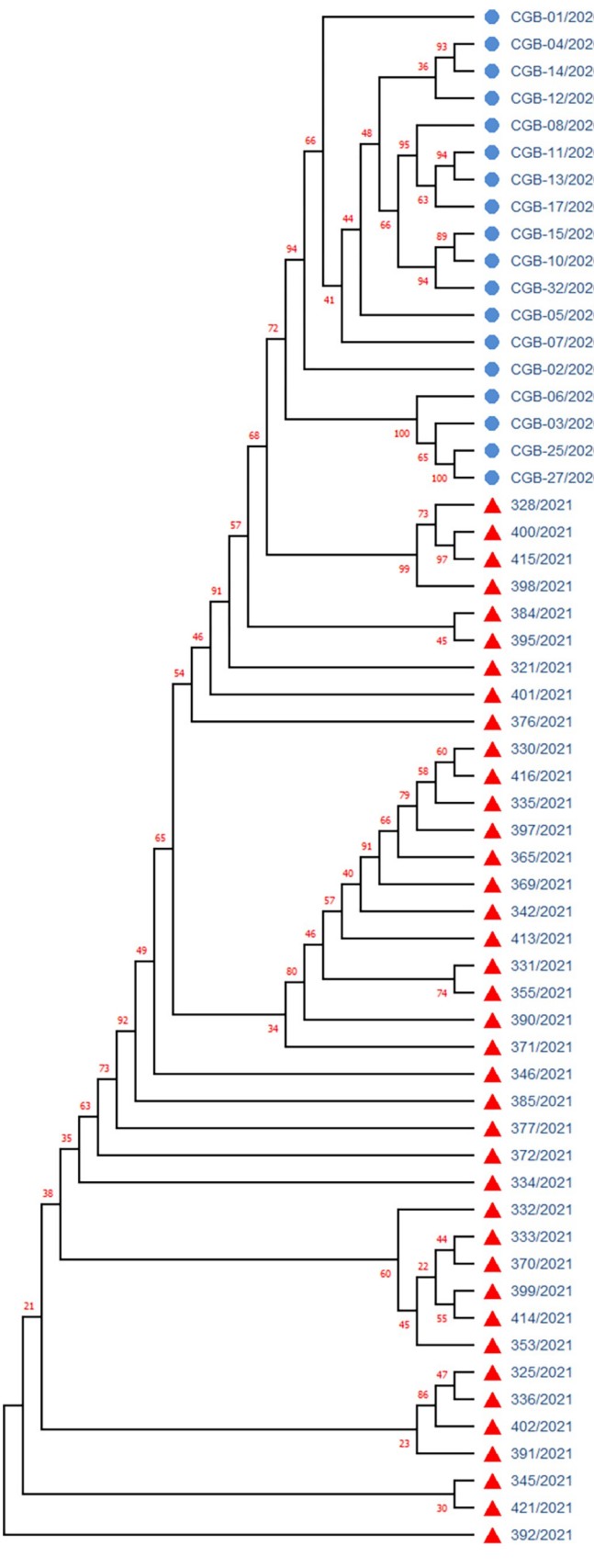

**Fig 13. Phylogenetic tree of the SARS-CoV-2 isolates distributed in Uzbekistan during the 1-st (2020) and 2-nd (2021) coronavirus pandemic waves.** Samples with blue circles are SARS-CoV-2 isolates from the first wave (November 2020), sample with red triangles are from the second wave (mid 2021).

The comparative study of the first wave virus genotypes (Fig 13) implied that the SARS-CoV-2 was distributed to Uzbekistan and not happened directly from China or Wuhan variant but has been spread through other countries after, at least, the first cycle of infection from entered countries. Sequencing based phylogeographic analysis in this study also suggested that there were the multiple independent viral introductions into Uzbekistan from North America, Europe, Africa and Asia [14], which were supported by the evidence of clustered outbreak/community transmission. The second peak occurred in middle of 2021 with the spread of the Delta variant, which eventually became dominant in the world [29] and later in Uzbekistan as shown in our current study (Figs 6 and 7). We suggest that the second wave of the SARS-CoV-2 genotype introduction have been mostly occurred through Russia and Turkey due to the strong socio-economic relationships between Uzbekistan and these countries.

The WGS of isolates our efforts have revealed that an average content of nucleotides was mainly consistent with reference SARS-CoV-2 genome (NC_045512) with slight reduction of G (0.012%) and C (0.051%) nucleotides in comparison to the NC_045512 where G and C accounted for 19.61% and 18.37% respectively [8]. It was found that RNA strand with a high number of C and G bases would form more stable stem-loops than that of a high number of T and A bases. This suggested that the SARS-CoV-2 is more efficient in reproduction than other coronaviruses because less energy is consumed in disrupting the secondary structures formed by its genomic RNA [8]. Because the Delta variant has a higher transmission rate compared to the previous variants, our data suggest that slight reduction of G and C content rate is an evolutionary adaptation of Delta variants for the rapid replication and reproduction in host population.

In our study, among the 12 classes of base substitutions, the C→T transition was dominant (42%) (Table 2; Fig 4). Abundance of C→T transition (40.6%) was observed among the first-wave SARAS-CoV-2 sequencing study from Uzbekistan [14]. Interestingly, we found asymmetry in base changes. For example, the rate of the C→T transition was much higher than that of its reverse T→C substitution (42% and 8%, respectively) (Table 2; Fig 4). Likewise, the rate of G→T transversion was five times higher than that of T→G substitution (16% and 3%, respectively) (Table 2; Fig 4). In this context, our results slightly less percentage from those reported previously by Yi et al. [30], but we observed similar pattern. Linear changes in the base composition over the time of spread were also observed in Ebola and influenza viruses [31]. Thus, the C→T and G→T asymmetry in the SARS-CoV-2 mutation spectra may be a characteristic of zoonotic RNA viruses recently introduced to human tissues [30].

Each of novel substitutions in the Nsp13: A85P (G16489C) and Nsp12: Y479N (T14875A) (Table 4), found in samples belonging to Delta variant's subclade 21A (GISAID sample IDs: EPI_ISL_3188967, EPI_ISL_3188979), has been reported for the first time in this study. Both substitutions are located in nonstructural proteins encoded by the ORF1b of a SARS-CoV-2 genome. Nsp13 is a superfamily 1 helicase which acts as motor protein that unwind a double-stranded nucleic acid into the two single-stranded nucleic acid. Nsp12 of coronaviruses encodes RNA-dependent RNA polymerase (RdRp) involved in replication of their genome and for the transcription of their genes [32]. Therefore, novel mutations found in these regions of the SARS-Cov-2 may play an important role in understanding of the COVID-19 disease epidemic.

In the beginning of 2021, the Alpha variant of SARS-CoV-2 was spread rapidly and became dominant in Uzbekistan (unpublished). The situation has changed after the rapid spread of the Delta variant worldwide. In current study, we observed that the Delta variant accounted for 82% among analyzed isolates and became dominant in Uzbekistan in the middle of 2021 (Fig 7). Recently, Nextstrain updated its clade designation, partitioning the 21A into the two subclades of 21J and 21I. Clade 21J includes the Delta variant, but it possesses additional ORF1a mutations such as A1306S, P2046L, P2287S, V2930L, T3255I as well as the ORF7b mutation T40I, and the N gene mutation G215C. Among analyzed Delta isolates in our data-set, 44% was designated to 21J subclade, whereas the rest belonged to 21A (56%) (Fig 7). There were no samples grouped into 21I subclade in our samples. According to Nextstrain database the Delta 21J variant became predominant worldwide (https://nextstrain.org/ncov/gisaid/global, accessed in December 2021).

The co-occurrence of substitutions in the N: R385K, ORF1a: S944L, H2092Y, L3644F, and ORF1b: H2285Y, identified in our study probably a sign of evolutionary divergence within the subclade 21A; therefore, these mutations require an attention and should be tracked in further studies. In the S gene, the T95I has occurred in 22% of Delta samples. This finding is in full concordance with previously reported study based on the analysis of 1276 Delta isolates [33], on the other hand, in the same study a distinct substitution G215C in the N genes was found as the Delta plus variants in 58% sequences samples, whereas in current study this substitution was identified in 46.8% of Delta isolates.

Analysis of predicted Nsp12 and Nsp13 stability due to the respective Tyr479Asn and Ala85Pro substitutions resulted in decrease of an average ΔΔG values of these proteins for minus 1.5; and minus 0.56, respectively (Fig 11). Decreased stability and low substitution frequency in these core proteins of the SARS-CoV-2, found in our study, suggested that the Nsp13:A85P and Nsp12:Y479N mutations may not have an evolutionary advantage for SARS-CoV-2. Nevertheless, it was reported that the NSP13 downregulates interferon production and signaling as well as NF-κB promoter signaling by limiting the TBK1 and IRF3 activation [34, 35], whereas the Nsp12 attenuates type I interferon production by inhibiting IRF3 nuclear translocation [36]. These implies that the Nsp13:A85P and Nsp12:Y479N substitutions may have an impact on primary interferon suppression in the host cells and can antagonize host antiviral innate immunity. Therefore, further research and monitoring of the spread of these mutations is required.

The statistically significant gender distribution bias among the subclades 21A and 21J (Fig 10), found in sequenced WGS in our study, could be a population specific gender susceptibility to the SARS-CoV-2 clades and underlies a need for further comparative studies in other populations. Previous global GISAID-derived metadata analysis also reported statistically significant gender bias among several SARS-CoV-2 clades [37]. Thus, there is a gender specific susceptibility to SARS-CoV-2 variants worldwide and further studies needed to be conducted to elucidate the possible genetic mechanisms of this phenomenon.

The association of ORF7a: P45L mutation with disease severity in Uzbekistan, reported for the first time herein (Table 10), is in concordance with previous research, that the stabilizing mutation in the ORF7a of a SARS-CoV-2 was associated with the increased severity and lethality in a group of Romanian patients, despite a lower viral copy number and a lower number of associated comorbidities [38].

The ORF7a protein is thought to be a type I transmembrane protein. The structure of the SARS-CoV protein ORF7a, shows similarities to the immunoglobulin-like (Ig-like) fold with some features resembling those of the Dl domain of ICAM-1 and suggests a binding activity to integrin I domains [39]. It is known that Ig-like domain-containing proteins play vital roles in mediating macromolecular interactions in the immune system. Since SARS-CoV-2 ORF7a,

similar to SARS-CoV ORF7a, it is predicted to be a member of the Ig-like domain superfamily [40–42]. ORF7a may play a significant role in the clinical severity of COVID-19 [43]. Recent study demonstrated that SARS-CoV-2 ORF7a coincubation with CD14+ monocytes *ex vivo* triggered a decrease in HLA-DR/DP/DQ expression levels and upregulated significant production of proinflammatory cytokines, including IL-6, IL-1β, IL-8, and TNF-α. Thus, it demonstrates that the SARS-CoV-2 ORF7a is an immunomodulating factor for immune cell binding and triggers dramatic inflammatory responses [42]. This forms the basis of a likely mechanism through which ORF7a mediates the potentially fatal cytokine storm progression in COVID-19 patients, indicating that ORF7a may be a key viral factor for disease severity.

The ORF7a: P45L mutation of the Delta became dominant in Russia [44]. It was also found in the Delta variants isolates from India [45], and in the Omicron samples from South Africa (EPI_ISL_6647958). We suggest that the 45L mutation in the ORF7a is the next step in the evolution of the coronavirus towards a decrease in the severity of the disease, since the main guarantee of the existence and spread of the virus is not the death of the host, but the evolution towards an increase in infectivity.

The study was limited by a relatively small number of samples that were subjected to WGS, since during the pandemic there were logistical issues with the timely supply of reagents in the required quantities. Another limitation of the study was the lack of SARS-CoV-2 samples from patients with asymptomatic disease, since such people do not usually seek medical help. Despite the limitations, the data obtained are valuable for understanding the spread and evolutionary features of SARS-CoV-2 specifically in Central Asian region. Further studies aimed at monitoring of SARS-CoV-2 variants distribution and their genetic diversity in the region are needed.

## Supporting information

**S1 File. Genome sequences of SARS-CoV-2 samples from Uzbekistan collected during the second pandemic wave.**
(FASTA)

**S2 File. Annotated genome variants of SARS-CoV-2 samples from Uzbekistan collected during second the pandemic wave.** Nucleotide and amino acid positions are indicated relative to the reference genome NC_045512.2 (MN908947).
(CSV)

## Acknowledgments

We thank the doctors and nurses of the COVID-19 clinics at the State Hospital Zangiota-1 Tashkent, Uzbekistan for their help in collection of the samples and disease data from symptomatic patients.

## Author Contributions

**Conceptualization:** Alisher Abdullaev, Shahlo Turdikulova.

**Data curation:** Alisher Abdullaev.

**Funding acquisition:** Dilbar Dalimova, Shahlo Turdikulova.

**Investigation:** Alisher Abdullaev, Abrorjon Abdurakhimov, Zebinisa Mirakbarova, Shakhnoza Ibragimova, Vladimir Tsoy, Sharofiddin Nuriddinov.

**Methodology:** Alisher Abdullaev, Zebinisa Mirakbarova, Dilbar Dalimova, Shahlo Turdikulova.

**Supervision:** Shahlo Turdikulova, Ibrokhim Abdurakhmonov.

**Visualization:** Alisher Abdullaev.

**Writing – original draft:** Alisher Abdullaev.

**Writing – review & editing:** Ibrokhim Abdurakhmonov.

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
