## [Decision Letter · Decision Letter 0]

3 May 2022

PONE-D-22-07071Genome sequence diversity of SARS-CoV-2 obtained from clinical samples in UzbekistanPLOS ONE

Dear Dr. Abdullaev,

Thank you for submitting your manuscript to PLOS ONE. After careful consideration, we feel that it has merit but does not fully meet PLOS ONE’s publication criteria as it currently stands. Therefore, we invite you to submit a revised version of the manuscript that addresses the points raised during the review process.

We look forward to receiving your revised manuscript.

Kind regards,

Vladimir Makarenkov

Academic Editor

PLOS ONE

Journal Requirements:

2. We noted in your submission details that a portion of your manuscript may have been presented or published elsewhere. 

(PLOS Genetics, Manuscript ID: PGENETICS-D-22-00251)

Please clarify whether this publication was peer-reviewed and formally published. If this work was previously peer-reviewed and published, in the cover letter please provide the reason that this work does not constitute dual publication and should be included in the current manuscript."

4. Please amend your authorship list in your manuscript file to include author  Abrorjon Abdurakhimov.

5. Thank you for stating in your Funding Statement: 

(This study has been supported by the research grant from the Ministry of Innovative Development, Republic of Uzbekistan (Research Grant number: А-ИРВ-2021-125). The funders had no role in study design, data collection and analysis, decision to publish, or preparation of the manuscript.)

Additional Editor Comments:

This paper is relevant and well written.

I think the authors could add a short discussion (preferably to the Introduction section) about possible origins of SARS-Cov-2. The references that could be cited are as follows:

Boni, Maciej F., et al. "Evolutionary origins of the SARS-CoV-2 sarbecovirus lineage responsible for the COVID-19 pandemic." Nature microbiology 5.11 (2020): 1408-1417.

Makarenkov, V., Mazoure, B., Rabusseau, G. et al. Horizontal gene transfer and recombination analysis of SARS-CoV-2 genes helps discover its close relatives and shed light on its origin. BMC Ecol Evo 21, 5 (2021).

Domingo JL. What we know and what we need to know about the origin of SARS-CoV-2. Environ Res. 2021;200:111785.

Moreover, the authors can also use a new SimPlot++ tool designed to detect recombination and visualize data using sequence similarity networks:

Samson, S. et al. SimPlot ++: a Python application for representing sequence similarity and detecting recombination, Bioinformatics, 2022; btac287 (https://academic.oup.com/bioinformatics/advance-article/doi/10.1093/bioinformatics/btac287/6572334?guestAccessKey=d079b57c-5b8e-4bf4-a1d6-06274bd89169).

Reviewers' comments:

Reviewer's Responses to Questions

**Comments to the Author**

1. Is the manuscript technically sound, and do the data support the conclusions?

Reviewer #1: Yes

Reviewer #2: Yes

2. Has the statistical analysis been performed appropriately and rigorously? 

Reviewer #1: Yes

Reviewer #2: Yes

3. Have the authors made all data underlying the findings in their manuscript fully available?

Reviewer #1: Yes

Reviewer #2: Yes

4. Is the manuscript presented in an intelligible fashion and written in standard English?

Reviewer #1: Yes

Reviewer #2: Yes

5. Review Comments to the Author

Reviewer #1: The manuscript describes genetic diversity of SARS-CoV-2 during the second wave of infection in Uzbekistan.

Despite, limited number of samples, authors found unique and consistent mutations that were related to major mutations at that time.

Manuscript shows interesting findings, e.g.,effect of amino acid substitution on the protein stability and disease severity. it would be an advantage if authors could elaborate little bit more on the relationship between amino acid substitution and disease severity.

There are some spelling an types which must be corrected before the final publication. Maybe the figure resolution is better in original file, but in the pdf I read it was impossible to read x/y axes labels, and color codes. please correct n=? in abstract.

should be improved.

It would be great if authors refer to their findings/figures, when they draw concluding statements in discussion (like in lines 435-438).

A figure providing an overview of similarity or dissimilarity of WGS of novel mutations Nsp13 and Nsp12 with other mutations reported in the databases would be useful to include.

Manuscript can be accepted after minor revision.

Reviewer #2: The manuscript from Abdullaev et al. describes the SARS-Cov-2 variability in Uzbek population from Tashkent. The samples were collected in the second wave of pandemic, when the Delta clade variants represented the majority. The whole-genome sequencing of the 35 viral samples provides information about virus variability in Uzbekistan at that time. Additional analyses of the patients affected by the virus provide an insight into relation of such variants to disease severity and sex ratio of affected individuals. The authors provide phylogenetic and variant analyses of the sequences with their additional structural characterisation and prediction of function for the novel ones.

Overall, the proposed study is well-done and presents a thoughtful investigation of the SARS-Cov-2 genetic variability and its impact on the COVID-19 infection on a sample from Uzbek population.

The manuscript would benefit from some additional English proofing to improve clarity of the messages. I have made only a few suggestions on the Pdf file with the text to help authors with minor pitfalls (see the attachment).

The abstract would need a conclusion phrase.

There should be more clarity about the comparison groups for the statistical association tests, especially those which appear to show significant difference, to make sure, what feature has been compared to what.

Since the Materials section is after the results section, some information about tools and methods used to obtain results would be helpful to the readers, such as:

“Using text X (or using method KKK, or software VVV) we compared groups NNN and MMM and detected YYY”.

It would make sense to have a final one / two phrase(s) of discussion being conclusive of the manuscript. The final paragraph of discussion finishes the manuscript abruptly.

It would be important to mention the limitation of the study closer to the end of discussion.

Minor comments:

Table 1 – there should probably be a median of the observed number of substitutions. It seems that this table could look better if transposed, so lines would become columns and vice versa.

Table 2 – the last column should better be in % and have two non-zero decimal points, like 42.12% or 42.0095% There should be the bottom line with the Total, where the % would sum up to 100% - is it like that now already? If the presented do not sum up to 100%, authors could another row showing the remaining %-ge and explaining, where it belongs to.

What is “aa position”? Amino acid? Please, spell out. What is “nt position”? Nucleotide?

Discussion

Page 29, line386 – Avoid such start of a part “So far, during the pandemic, several factors…”. Instead, authors may use “The growth of the number of individuals affected by COVID-19 has been affected…”

All numbers smaller than 10 should be spelled out.

No new figures should ne presented in the discussion. Fig 13 should first be presented in the intro.

Figure 1 Please, add the axis name, such as “Date”. Y axis, there should be the text of axis name, as proposed but with addition of “, n individuals”

Figure 2 same as F1.

Rest of figures – all axes should have a name, where there are groupings, the authpors should explain, what they mean in the legends, like “nt_substitutions” – spell out the meaning. Figures 3+ for Y axis, the name should have “events” in plural. Figures with Mutation events might benefit from a violin plot style presentation as compared to the box plot, given the relatively small N of events and genomes in the study.

The phylogeny figure 6 is of insufficient image quality. Impossible to read the detail. Could authors add the corona plot to the supplementary, if any.

Figure 9 should be also presenting the age by sex.

6. PLOS authors have the option to publish the peer review history of their article (what does this mean?). If published, this will include your full peer review and any attached files.

Reviewer #1: No

Reviewer #2: No

---

## [Author Response · Author response to Decision Letter 0]

7 Jun 2022

Response to Academis Editor:

Dear editor, thank you for valuable comments.

1. We have formatted the manuscript according to PLOS ONE's style requirements, we also have changed the order of sections, materials and methods have been moved after the introduction (see revised manuscript with track changes).

The figure files have been corrected by PACE to meet PLOS ONE requirements

2. The manuscript initially was submitted to PLOS Genetics (Manuscript ID: PGENETICS-D-22-00251) but not considered for publication by editors. Although the editors at that journal were not able to consider manuscript for publication, they encouraged us to transfer the manuscript to another PLOS journal. They have made this recommendation based on their assessment of manuscript, their knowledge of the other journal, and after consultation with other journal's editors. So, we state that the manuscript was not peer-reviewed and/or formally published elsewhere and has not been submitted simultaneously for publication elsewhere.

3. We provided repository information during submission, please hold it until acceptance. We do not wish to make changes to our Data Availability statement.

4. Thank you, we amended authorship list.

5. We will amend updated Funding Statement as follows: “This study has been supported by the research grant from the Ministry of Innovative Development, Republic of Uzbekistan (Research Grant number: А-ИРВ-2021-125). There was no additional external or internal funding received for this study. The funders had no role in study design, data collection and analysis, decision to publish, or preparation of the manuscript”. We included our amended Funding Statement within second cover letter.

6. We have updated preprints to published versions (if available) and corrected the order of references cited in the manuscript. We added one new reference and removed one (see revised manuscript with track changes)

7. According to your recommendation, a short information about possible origin of SARS-CoV-2 (citing Makarenkov et al., 2021) was added in the Introduction.

8. Thank you for suggesting a new SimPlot++ tool. We installed Windows version of this software and performed several runs. It is very useful program, but we need time to figure out which parameters suits better for analysis and how to interpret the results. For sure, we will use it to present further results, as work on collecting genomic information of SARS-CoV-2 in Uzbekistan still in progress.

Responce to Reviewer #1:

Dear reviewer, we appreciate the time you spent to review our manuscript and gave valuable recommendations

1. Amino acid substitution might change protein stability, thus might increase or decrease its functional properties. In the discussion section we tried to explain possible mechanisms of amino acid substitution in ORF7a which could be a mediator of proinflammatory cytokines production and triggers dramatic inflammatory responses resulted in disease severity.

2. N= corrected, missing value added

3. Original pictures submitted to the Journal are in good quality, after compression to pdf. the resolution is drastically reduced. I think the editor can send you the original image files by request.

4. Thank you for this suggestion, since we are continuously getting more genomic data from new samples, in our next study we will provide a detailed overview of dissimilarity of WGS of novel mutations with other mutations reported in the databases.

5. We have added references to our findings/figures in еру Discussion section as you suggested

Responce to Reviewer #2:

Dear reviewer, we appreciate the time you spent to review our manuscript and gave valuable recommendations

1. We have corrected the manuscript according to your suggestions on the Pdf file

2. Conclusion phrase have been added to abstract.

3. We have made corrections in groups comparison, according to your recommendations

4. We have updated information in the Results where it was appropriate 

5. The limitation has been added closer to the end of discussions as you suggested

6. The Table 1 has been transposed

7. The Table 2 has been corrected 

8. The abbreviations “aa” (amino acid) and “nt” (nucleotide) have been spelled out in the text

9. The phrase “So far, during the pandemic, several factors…”. has been changed as you recommended

10. Corrected. All number smaller than has been 10 spelled out

11. We believe that Fig 13 is more suitable for discussion in its content

12. Figure 1, suggested axis names have been added

13. Figure 2, suggested axis names have been added

14. Rest figures have been corrected according to your recommendations

15. Figures with Mutation events have been changed to Violin plot style instead of Box plot

16. The image has a highest possible resolution generated by the Nexclade. The quality is better in original tif., but reduced in pdf.

17. We do not have the corona plot

---

## [Editor Report · Decision Letter 1]

8 Jun 2022

Genome sequence diversity of SARS-CoV-2 obtained from clinical samples in Uzbekistan

PONE-D-22-07071R1

Dear Dr. Abdullaev,

We’re pleased to inform you that your manuscript has been judged scientifically suitable for publication and will be formally accepted for publication once it meets all outstanding technical requirements.

Kind regards,

Vladimir Makarenkov

Academic Editor

PLOS ONE
---

## [Editor Report · Acceptance letter]

14 Jun 2022

PONE-D-22-07071R1 

Genome sequence diversity of SARS-CoV-2 obtained from clinical samples in Uzbekistan 

Dear Dr. Abdullaev:

I'm pleased to inform you that your manuscript has been deemed suitable for publication in PLOS ONE. Congratulations! Your manuscript is now with our production department. 

Kind regards, 

on behalf of

Dr. Vladimir Makarenkov 

Academic Editor

PLOS ONE